

# SedFoam-2.0: a 3D two-phase flow numerical model for sediment transport

Julien Chauchat[1], Zhen Cheng[2,3], Tim Nagel[1], Cyrille Bonamy[1], and Tian-Jian Hsu[2]

[1]University of Grenoble Alpes, LEGI, G-INP, CNRS, F-38000 Grenoble, France
[2]Civil and Environmental Engineering, Center for Applied Coastal Research, University of Delaware, Newark, DE 19711, USA
[3]Now at Applied Ocean Physics & Engineering, Woods Hole Oceanographic Institution, Woods Hole, MA 02543, USA

*Correspondence to:* julien CHAUCHAT (julien.chauchat@univ-grenoble-alpes.fr)

**Abstract.**

In this paper, a three-dimensional two-phase flow solver, SedFoam-2.0, is presented for sediment transport applications. The solver is extended upon twoPhaseEulerFoam available in the 2.1.0 release of the open-source CFD toolbox OpenFOAM. In this approach the sediment phase is modeled as a continuum, and constitutive laws have to be prescribed for the sediment stresses.

In the proposed solver, two different inter-granular stress models are implemented: the kinetic theory of granular flows and the dense granular flow rheology $\mu(I)$. For the fluid stress, laminar or turbulent flow regimes can be simulated and three different turbulence models are available for sediment transport: a simple mixing length model (one-dimensional configuration only), a $k - \varepsilon$ and a $k - \omega$ model. The numerical implementation is first demonstrated by two validation test cases, sedimentation of suspended particles and laminar bed-load. Two applications are then investigated to illustrate the capabilities of SedFoam-2.0 to

deal with complex turbulent sediment transport problems with different combinations of inter-granular stress and turbulence models.

## 1 Introduction

Sediment transport is the main process that drives the morphological evolution of fluvial and coastal environments. Consequently, the ability to predict sediment transport is a major societal issue for the management of natural systems in order

to limit and prevent the impacts related to extreme events exacerbated by climate change and human activities such as construction of hard structures (dams, harbors, dikes, *etc.*), land reclamation, and dredging. Addressing these issues requires the development of comprehensive models that account for the variety of complex hydrosedimentary processes such as particle interactions with hydrodynamic and flow turbulence or particle-particle interactions due to collisions or frictions. However, these complex phenomena are only poorly understood at present and they are incompletely integrated into engineering tools

to predict the coastal and river morphodynamic As a result, our prediction performance is limited. Improving these models is urgently needed by land settlement decision-makers for the management of water resources and environmental issues. The development of more accurate sediment transport models integrating the complexity of the underlying physical coupling mechanisms is the main goal of the community model presented herein.



The processes at work in sediment transport are numerous. In classical sediment transport definitions, particles can be transported as suspended-load, *i.e.* without contact with the sediment bed, or as bed-load, *i.e.* with permanent or intermittent contact with the sediment bed by rolling, sliding or saltation (Fredsoe and Deigaard, 1992). In the suspended-load, sediment concentration is quite low and the sediment suspension is driven by their interactions with turbulent eddies. In the near bed

region, the sediment concentration increases drastically and can reach values as high as ∼60% in volume fraction. As a result, the particle-particle interactions such as collisions or frictional contacts become dominant. When the granular particles are agitated by a strong shear stress, a thick layer of particles are mobilized and transported above the static bed. This regime, the so-called sheet flow regime, can significantly contribute to the sediment transport and the morphological evolution of rivers, for example under extreme flood conditions (Hanes and Bowen, 1985), and in the coastal zone, especially in the surf zone and

swash zone (e.g., Lanckriet et al., 2014; Aagaard et al., 2002; van der A et al., 2013).

From the modeling perspective, the classical modeling approach consists in dividing the physical domain into two sublayers. The upper layer corresponds to the water column in which depth-integrated or depth resolving Reynolds Averaged Navier-Stokes (RANS) equations are solved and the sediment concentration is assumed to be dilute in which the sediment particles are treated as passive scalar with a settling velocity difference with the fluid phase. The lower near-bottom bed-load layer is solved

by using a bed shear stress based empirical formulas for the sediment flux (e.g., Meyer-Peter and Muller, 1948) coupled with the sediment mass conservation equation (Exner equation, Exner, 1925). The latter allows to predict the bed morphological evolution. These two layers are dynamically coupled using empirical sediment vertical fluxes: deposition and pick-up fluxes (Van Rijn, 1984), which are also parameterized based on the bed shear stress. The single phase model has been widely used because of its simplicity and computational efficiency, and it has been integrated into meso/large scale models, such as Delft3D

(Lesser et al., 2004; Hu et al., 2009), XBeach (Roelvink et al., 2009) and ROMS (Warner et al., 2008). Despite their ability to predict long-term regional morphology in littoral zones, there are several challenges to model sediment transport using such single-phase methodology. Firstly, major transport in sheet flow occurs within about $20 \sim 50$ grain diameters above the bed. Hence, the resolved suspended load layer only accounts for a minor portion of transport while the majority of the predicted transport relies on the empirical bedload parameterization. Secondly, most of the existing formulae for bedload transport and

suspension flux are developed for steady flow and hence their applicability to the highly unsteady wave-induced transport is questionable (Yu et al., 2012). Thirdly, the particle-particle collisions and frictions in the high concentration regions are assumed to be negligible, and entrained sediments are assumed to be suspended immediately by the flow turbulence. Fourthly, the bed erosion/deposition and its impact on the sediment concentration distribution and the carrier flow fields are largely neglected. In addition, simple parametrization solely on the bed shear stress may be insufficient, for example, the role of

pressure gradient in sediment transport has been identified for extreme events such as storms (Foster et al., 2006; Cheng et al., 2017). These assumptions strongly limit the capability of such single-phase model, thus the development of more advanced numerical models are motivated for sediment transport applications.

During the past two decades, an increasing amount of research efforts are devoted to develop two-phase flow models for sediment transport (see a brief summary in table 1). In this two-phase flow approach, dynamical equations are solved for both

the fluid phase (water) and the particle phase (sediment), with the latter being seen as a continuous phase dispersed in the



**Table 1.** Summary of Eulerian two-phase models for sediment transport applications

| Authors | Turbulence model | Particle stress |
|---|---|---|
| Asano (1990); Li and Sawamoto (1995); Dong and Zhang (2002) | mixing length | Bagnold |
| Jenkins and Hanes (1998) | mixing length | kinetic theory |
| Revil-Baudard and Chauchat (2013) | mixing length | granular rheology |
| Li et al. (2008) | $k-$L | Bagnold |
| Bakhtyar et al. (2009) | $k-\varepsilon$ | Bagnold |
| Hsu and Liu (2004); Chauchat and Guillou (2008); Amoudry et al. (2008) Yu et al. (2010); Cheng et al. (2017) | $k-\varepsilon$ | kinetic theory |
| Amoudry (2014); Jha and Bombardelli (2009, 2010) | $k-\omega$ | kinetic theory |
| Lee et al. (2016) | $k-\varepsilon$ | granular rheology |

fluid. The two-phase flow approach gives a general modeling framework that potentially allows to take into account almost all the physical processes involved in sediment transport, such as fluid-particle interactions, turbulence modulation, and particle-particle interactions. From a technical point of view, the difficulty is in solving two non-linearly coupled Navier-Stokes type of partial differential equations. From a theoretical perspective, it is difficult to incorporate various physical processes that take

place at smaller scales than the averaging scale, which is used to derive the two-phase equations. With closures for particle-particle interactions, flow turbulence and turbulence-sediment interactions, the Eulerian two-phase model excels the single phase sediment transport models in several aspects, and provide us new insights into sediment transport mechanisms.

The key closures in the two-phase flow sediment transport models are flow turbulence and granular stresses closures. In terms of the turbulence closure, the first one that has been tested was a mixing length model by Jenkins and Hanes (1998)

followed by others (e.g. Dong and Zhang, 1999; Revil-Baudard and Chauchat, 2013). The other turbulence model that has been used is the $k-\varepsilon$ model (e.g. Hsu et al., 2003; Longo, 2005; Bakhtyar et al., 2009). Reynolds stress model has been test by Jha and Bombardelli (2010) and more recently $k-\omega$ model has been tested by Jha and Bombardelli (2009) and Amoudry (2014). Concerning the granular stress models, the first model that has been tested by Hanes and Bowen (1985) is the empirical rheology of Bagnold (1954). This rheology has then been used by Dong and Zhang (1999); Bakhtyar et al. (2009) for oscillatory

sheet flow applications. Jenkins and Hanes (1998) were the first to apply the kinetic theory of dense granular flows in a two-phase flow model for sediment transport. The kinetic theory has been used quite extensively by different authors to study sediment transport (e.g. Hsu and Liu, 2004; Berzi, 2011; Berzi and Fraccarollo, 2013; Cheng et al., 2017). More recently, the dense granular flow rheology initially proposed by GDRmidi (2004) for dry granular flows has been used for sediment transport applications in the laminar flow regime by Ouriemi et al. (2009) and later to turbulent flow conditions by Revil-

Baudard and Chauchat (2013),Chiodi et al. (2014) and Lee et al. (2016). Due to the complexity of the model formulation, most of the existing two-phase models are based on the Reynolds-averaged approach and simplified into one-dimensional form (e.g. Hanes and Bowen, 1985; Jenkins and Hanes, 1998; Hsu et al., 2003; Revil-Baudard and Chauchat, 2013), with a few exceptions





(two-dimensional models, Chauchat and Guillou, 2008; Bakhtyar et al., 2009; Amoudry and Liu, 2009). Only very recently, three dimensional (3D) large-eddy simulation two-phase flow model has been applied for sheet flow sediment transport (Cheng et al., submitted). These 3D numerical models are based on the open-source CFD toolbox openFoam.

The purpose of the present contribution is to follow up on Cheng et al. (2017) work by adding new capabilities to the open-
source model sedFoam. In particular, the mixing length turbulence model and dense granular flow rheology used by Revil-Baudard and Chauchat (2013) and Chauchat (2017) for sheet flows have been implemented. In addition, we implemented and tested the $k - \omega$ turbulence model for two-phase flow sediment transport modeling purposes. Our final goal is to provide a comprehensive numerical framework that solves the two-phase flow equations in three dimensions with the capability to select different combinations of turbulent model and granular stress model for sediment transport applications. By disseminating the
numerical model in the open-source framework, in the long run, we expect new capabilities will be added to the model by the scientific community. We strongly believe that developing such an open-source community model is the only effective way to make significant progress.

The paper is organized as follows, in section 2 the mathematical formulation of the model is presented together with the closures for drag, turbulence model and granular stress models. In section 3, the semi-discretized form of the equations and
the velocity-pressure coupling algorithm are presented in detail. In section 4, two different one-dimensional test cases are presented, namely sedimentation and laminar bed load for which experimental data or analytical solutions exist. In order to illustrate the capabilities of the numerical model for more complex sheet flow, and multi-dimensional problems, SedFoam-2.0  is used to simulate unidirectional turbulent sheet flow and the scour at an apron in two-dimensions in section 5. Finally, summaries and conclusions are drawn in section 6.

## 2  Mathematical formulation

The mathematical formulation of the Eulerian two-phase flow model is obtained by averaging local and instantaneous mass and momentum conservation equations over fluid and dispersed particles. Different averaging operators can be used, ensemble averaging (Drew, 1983) or spatial averaging (Jackson, 2000), and provided that the mathematical derivation is done properly the different approaches should lead to the same conservation equations (Zhang and Prosperetti, 1997; Jackson, 1997). The
resulting governing equations can be considered as the counterpart of the clear fluid Navier-Stokes equations for single phase flow. In order to apply these equations to turbulent flow, in which turbulent motions are generated by flow shear much larger than the grain scale, additional turbulence averaging or filtering is required. In the present model, turbulence-averaged Eulerian two-phase flow equations are derived by following a similar procedure presented in Hsu et al. (2003); Hsu and Liu (2004).

### 2.1  Turbulence-averaged two-phase flow governing equations

The mass conservation equations for the particle phase and fluid phase are written as:

$$\frac{\partial \alpha}{\partial t} + \frac{\partial \alpha u_i^a}{\partial x_i} = 0, \tag{1}$$





$$\frac{\partial \beta}{\partial t} + \frac{\partial \beta u_i^b}{\partial x_i} = 0, \tag{2}$$

where $\alpha$, and $\beta = 1 - \alpha$ are the particle and fluid volume concentrations, $u_i^a, u_i^b$ are the sediment and fluid phase velocities, and $i = 1, 2, 3$ represents streamwise, spanwise and vertical component, respectively. The momentum equations for fluid and particle phases can be written as:

$$\frac{\partial \rho^a \alpha u_i^a}{\partial t} + \frac{\partial \rho^a \alpha u_i^a u_j^a}{\partial x_j} = -\alpha \frac{\partial p}{\partial x_i} + \alpha f_i - \frac{\partial \tilde{p}^a}{\partial x_i} + \frac{\partial \tau_{ij}^a}{\partial x_j} + \alpha \rho^a g_i + \alpha \beta K (u_i^b - u_i^a) - S_{US} \, \beta K \nu_t^b \frac{\partial \alpha}{\partial x_i}, \tag{3}$$

$$\frac{\partial \rho^b \beta u_i^b}{\partial t} + \frac{\partial \rho^b \beta u_i^b u_j^b}{\partial x_j} = -\beta \frac{\partial p}{\partial x_i} + \beta f_i + \frac{\partial \tau_{ij}^b}{\partial x_j} + \beta \rho^b g_i - \alpha \beta K (u_i^b - u_i^a) + S_{US} \, \beta K \nu_t^b \frac{\partial \alpha}{\partial x_i}, \tag{4}$$

where $\rho^a, \rho^b$ are particle and fluid density, respectively, $g_i$ is the gravitational acceleration and $p$ is the fluid pressure. $f_i$ is the external force that drives the flow. The fluid stress $\tau_{ij}^b$ includes fluid grain-scale (viscous) stress and fluid Reynolds stresses (see section 2.2) and $\tilde{p}^a, \tau_{ij}^a$ are particle normal stress and shear stress (see section 2.3). The last two terms on the right-hand-side (RHS) of equations 3 and 4 are momentum coupling between the fluid phase and particle phase through drag force, where $K$ is the drag parameter. In particular the second to the last term represents averaged drag force due to mean relative velocity between fluid and particle phases, while the last term represents the fluid turbulent suspension term, also called drift velocity by Simonin (1991). This term is due to the correlation of sediment concentration and fluid velocity fluctuations and the gradient transport assumption is adopted here for its closure. Hence, $\nu_t^b$ is the turbulent viscosity to be calculated using a turbulence closure, and $S_{US} = 1/\sigma_c$ is inverse of the the Schmidt number. This term is equivalent to the turbulent suspension flux of the Rouse profile in the two-phase flow formalism (see Chauchat (2017) appendix 1 for a detailed demonstration).

The drag parameter $K$, is modeled following Schiller and Naumann (1933):

$$K = 0.75 C_d \frac{\rho^b}{d_{eff}} \parallel \mathbf{u^b} - \mathbf{u^a} \parallel \beta^{-h_{Exp}} \tag{5}$$

where $d_{eff} = \psi d$ is the effective sediment diameter, in which $\psi$ is the shape factor and $d$ is the particle diameter. The hindrance function $\beta^{-h_{Exp}}$ represents the drag increase when the particle volume concentration increases. $h_{Exp}$ is the hindrance exponent that depends on the particulate Reynolds number. For simplicity, the value of $h_{Exp}$ is assumed to be constant (default value is 2.65), and its value can be specified from the *constant/transportProperties* file in SedFoam-2.0. The drag coefficient $C_d$ is calculated as:

$$C_d = \begin{cases} \dfrac{24}{Re_p}(1 + 0.15 Re_p^{0.687}), & Re_p \leq 1000 \\ 0.44, & Re_p > 1000 \end{cases}, \tag{6}$$

in which, the particulate Reynolds number $Re_p$ is defined as: $Re_p = \beta \parallel \mathbf{u^b} - \mathbf{u^a} \parallel d_{eff}/\nu^b$, where $\nu^b$ stands for the fluid kinematic viscosity. This drag model can be chosen by the keyword "GidaspowSchillerNaumann" in the file *constant/interfacialProperties*. Even though other drag models are available, we omitted their descriptions as our major issue issue in developing a Eulerian two-phase flow model is to provide closure laws for turbulence closures and granular stress models. This will be extensively discussed in the following subsections, particularly different modeling options available in SedFoam-2.0 are presented.





## 2.2 Fluid phase shear stress

Because the present model equations are obtained by averaging over turbulence, the fluid stresses are consisted of a large-scale component $R_{ij}^{bt}$ (*i.e.*, Reynolds stress) and a grain-scale stress $r_{ij}^b$, which includes the viscous stress and an additional effect due to fluid-particle interaction at grain scale. The total fluid stress is written as:

$$\tau_{ij}^b = R_{ij}^{bt} + r_{ij}^b = \rho^b \beta \Big[ 2\nu_{Eff}^b \, S_{ij}^b - \frac{2}{3} k \delta_{ij} \Big], \tag{7}$$

5    where, $\nu_{Eff}^b = \nu_t^b + \nu^{mix}$ is the fluid phase effective viscosity with $\nu_t^b$ being the eddy viscosity, and $\nu^{mix}$ is the mixture viscosity. $S_{ij}^b$ is the deviatoric part of the fluid phase strain rate tensor defined as:

$$S_{ij}^b = \frac{1}{2} \Big( \frac{\partial u_i^b}{\partial x_j} + \frac{\partial u_j^b}{\partial x_i} \Big) - \frac{1}{3} \frac{\partial u_k^b}{\partial x_k} \delta_{ij}, \tag{8}$$

The Reynolds stress tensor $R_{ij}^{bt}$ is modeled as:

$$R_{ij}^{bt} = \rho^b \beta \Big[ 2\nu_t^b \, S_{ij}^b - \frac{2}{3} k \delta_{ij} \Big], \tag{9}$$

and the viscous stress tensor is modeled as:

$$r_{ij}^b = 2\rho^b \beta \nu^{mix} \, S_{ij}^b, \tag{10}$$

In SedFoam-2.0, several different viscosity or turbulence closures are implemented, and these models can be selected according to specific flow conditions ranging from laminar to turbulent flows, and in particular, the mixture viscosity can be selected in combination of granular rheology model for the granular stresses (see section 2.3.2). For the turbulent eddy viscosity, modified turbulence closures are implemented for sediment transport applications, and they are presented in subsection

15    2.2.2.

### 2.2.1 Mixture viscosity

The mixture viscosity model mostly depends on the particle phase volume concentration. Four different models are available in SedFoam-2.0. In the pure fluid model the mixture viscosity is equal to the fluid one: $\nu^{mix} = \nu^b$. This model is selected by default when using the kinetic theory of granular flows and can be selected by setting *none* for the keyword *FluidViscosityModel*

20    in the file *constant/granularRheologyProperties*.

The Einstein model (Einstein, 1906) is valid for very dilute situation ($\alpha < 0.01$):

$$\frac{\nu^{mix}}{\nu^b} = 1 + 2.5\alpha. \tag{11}$$

The phenomenological model proposed by Krieger and Dougherty (1959) is valid for very dense situations:

$$\frac{\nu^{mix}}{\nu^b} = \Big( 1 - \frac{\alpha}{\alpha_{max}} \Big)^{-n}, \tag{12}$$





where $\alpha_{max}$ is the maximum volume concentration and $n$ is an empirical exponent usually taken as $n = 2.5\alpha_{max}$ for consistency with Einstein's model at low volume concentration.

The model proposed by Boyer et al. (2011) is also empirical but has been obtained based on detailed rheological experiment. It is consistent with Enstein's model for low volume concentration and Krieger-Dougherty's model at high volume concentrations:

$$\frac{\nu^{mix}}{\nu^b} = 1 + 2.5\alpha \left( 1 - \frac{\alpha}{\alpha_{max}} \right)^{-1}.$$ (13)

The choice of mixture viscosity model is made in the file *constant/granularRheologyProperties* through the keyword *FluidViscosityModel* and is only available when the granular rheology is chosen.

**Table 2.** Fluid mixture viscosity model

| FluidViscosityModel | none | Einstein | KriegerDougherty | BoyerEtAl |
|---|---|---|---|---|
| $\nu^{mix}/\nu^b =$ | 1 | Eq. (11) | Eq. (12) | Eq. (13) |

A special treatment of the terms $1 - \alpha/\alpha_{max}$ is needed to avoid dividing by zero, basically, a clipping of $\alpha/\alpha_{max}$ is performed.

### 2.2.2 Turbulence modeling

As discussed above, the turbulence averaged formulation requires a closure for the eddy viscosity. Three turbulence models are available in SedFoam-2.0: a mixing length model (only valid for 1D configuration), the $k - \varepsilon$ model from Cheng and Hsu (2014); Cheng et al. (2017) and a $k - \omega$ turbulence model introduced in the present contribution. The turbulence model can be selected using the *RASModel* keywords in the file *constant/RASProperties* and specific parameters of the two-phase turbulence models are set in the file *constant/twophaseRASProperties* (see table 3).

**Table 3.** Fluid turbulence model

| turbulence model | laminar | mixing length (1D only) | $k - \varepsilon$ | $k - \omega$ |
|---|---|---|---|---|
| keywords | laminar | twophaseMixingLength | twophasekEpsilon | twophasekOmega |

#### 2.2.2.1 Laminar

For laminar flow applications, the turbulence model is turned off by setting $\nu_t^b = 0$, and $k = 0$, however, the mixture viscosity model can be selected to account for the sediment effect on the mixture viscosity, thus the effective fluid viscosity is calculated as: $\nu_{Eff}^b = \nu^{mix}$.





#### 2.2.2.2 Mixing length (1D only)

In the mixing length approach the eddy viscosity is modelled using a simple algebraic equation:

$$\nu_t{}^b = l_m^2 \parallel \nabla \mathbf{u_b} \parallel \tag{14}$$

$$l_m = \kappa \int\limits_0^y 1 - \left( \frac{\alpha(\xi)}{\alpha_{max}} \right)^{1.66} d\xi, \tag{15}$$

where $\kappa$ is the von Karman constant and the exponent 1.66 has been proposed by Chauchat (2017) based on matching experimental data from Revil-Baudard et al. (2015). This model is only working in 1D configuration for which the direction of gravity is $y$. This turbulence model has been implemented mostly for compatibility with earlier works.

#### 2.2.2.3 $k - \varepsilon$ model

Cheng et al. (2017) have implemented the $k-\varepsilon$ model refined from Hsu et al. (2004) and Yu et al. (2010), in which the turbulent eddy viscosity $\nu_t^b$ is calculated by :

$$\nu_t^b = C_\mu \frac{k^2}{\varepsilon}, \tag{16}$$

where $C_\mu$ is an empirical coefficient (see Table 4). The Turbulent Kinetic Energy (TKE) $k$ is computed from the solution of equation (17), appropriate for sand particles in water:

$$\frac{\partial k}{\partial t} + u_j^b \frac{\partial k}{\partial x_j} = \frac{R_{ij}^{bt}}{\rho^b} \frac{\partial u_i^b}{\partial x_j} + \frac{\partial}{\partial x_j} \left[ \left( \nu^b + \frac{\nu_t^b}{\sigma_k} \right) \frac{\partial k}{\partial x_j} \right] - \varepsilon - \frac{2K(1 - t_{mf})\alpha k}{\rho^b} - \frac{S_{US}}{\beta} \nu_t^b \frac{\partial \alpha}{\partial x_j} \left( \frac{\rho^a}{\rho^b} - 1 \right) g_j, \tag{17}$$

The above $k$-equation is similar to the clear fluid $k-\varepsilon$ closure, except that the last two terms on the RHS in Eq. (17) take account for the sediment damping effect on the carrier flow turbulence through drag (the fourth term) and density stratification(the last term). In the drag-induced damping term, the parameter $t_{mf}$ is introduced to characterize the degree of correlation between particles and fluid velocity fluctuations and it can be quantified by the Stokes number $St$ (Benavides and van Wachem, 2008):

$$St = \frac{t_p}{t_l}, \tag{18}$$

where $t_p = \rho^a/(\beta K)$ is the particle response time, $t_l = k/(6\varepsilon)$ is the characteristic time scale of energetic eddies. Danon et al. (1977) and Chen and Wood (1985) proposed an exponential function for $t_{mf}$, which is also used in (Cheng et al., 2017):

$$t_{mf} = e^{-B \cdot St}, \tag{19}$$

where $B$ is an empirical coefficient. The last term in the TKE equation (17) represents the buoyancy term. For typical sediment concentration with an upward decaying profile, this term represents the well-known sediment-induced stable density stratification that provide another source of turbulence attenuation.





Finally, the balance equation for the rate of turbulent kinetic energy dissipation $\varepsilon$ is written as:

$$\frac{\partial \varepsilon}{\partial t} + u_j^b \frac{\partial \varepsilon}{\partial x_j} = C_{1\varepsilon} \frac{\varepsilon}{k} \frac{R_{ij}^{bt}}{\rho^b} \frac{\partial u_i^f}{\partial x_j} + \frac{\partial}{\partial x_j} \left[ \left( \nu^b + \frac{\nu_t^b}{\sigma_\varepsilon} \right) \frac{\partial \varepsilon}{\partial x_j} \right] - C_{2\varepsilon} \frac{\varepsilon^2}{k} - C_{3\varepsilon} \frac{\varepsilon}{k} \frac{2K(1-t_{mf})\alpha k}{\rho^b} - C_{4\varepsilon} S_{US} \frac{\varepsilon}{k\beta} \nu_t^b \frac{\partial \alpha}{\partial x_j} \left( \frac{\rho^a}{\rho^b} - 1 \right) g_j \quad (20)$$

As discussed in Hsu et al. (2004), due to lack of comprehensive experimental data, the coefficients associated with the present two-equation closure are adopted from their clear fluid counterpart. The coefficient $C_{3\varepsilon}$ in the $\varepsilon$ equation (20) is chosen to be 1.2. For the coefficient associated with the buoyancy term, $C_{4\varepsilon} = 0$ is used in stably stratified condition, while it is set to

1 for unstably stratified condition. Table 4 summarizes the model coefficients. These coefficients were shown to work well for typical medium to coarse sand transport (Hsu et al., 2004; Yu et al., 2010; Cheng et al., 2017). Furthermore, it was found that the coefficient $B$ (see Eq. (19)) is sensitive to the model result and thus the coefficient $B$ is chosen to be a free parameter to be calibrated with measured data.

**Table 4.** $k - \varepsilon$ model coefficients.

| $C_\mu$ | $C_{1\varepsilon}$ | $C_{2\varepsilon}$ | $C_{3\varepsilon}$ | $C_{4\varepsilon}$ | $\sigma_k$ | $\sigma_\varepsilon$ | $S_{US}$ |
|---|---|---|---|---|---|---|---|
| 0.09 | 1.44 | 1.92 | 1.2 | 0 or 1 | 1.0 | 1.3 | 1 |

#### 2.2.2.4   $k - \omega$ **model**

Motivated by Amoudry (2014), a two-phase $k - \omega$ turbulence model is introduced in the present contribution. The turbulent eddy viscosity $\nu_t^b$ is calculated by:

$$\nu_t^b = \frac{k}{\omega} \tag{21}$$

Following the same method of developing the two-phase $k - \varepsilon$ turbulence model for sediment transport, the modification to the equations for the fluid TKE and the fluid specific rate of turbulent energy dissipation $\omega$ is made by adding the effect of the particle phase presence to the clear fluid $k - \omega$ model (i.e the particle drag and the buoyancy terms).

The fluid turbulent kinetic energy equation reads as:

$$\frac{\partial k}{\partial t} + u_j^b \frac{\partial k}{\partial x_j} = R_{ij}^{bt} \frac{\partial u_i^b}{\partial x_j} + \frac{\partial}{\partial x_j} \left[ \left( \nu^b + \frac{\nu_t^b}{\sigma_k} \right) \frac{\partial k}{\partial x_j} \right] - C_\mu k\omega - \frac{2K(1-t_{mf})\alpha k}{\rho^b} - \frac{S_{US}}{\beta} \nu_t^b \frac{\partial \alpha}{\partial x_j} \left( \frac{\rho^a}{\rho^b} - 1 \right) g_j. \tag{22}$$

Following the same reason used in the $k - \varepsilon$ closure, the equation for the fluid specific rate of turbulent energy dissipation $\omega$ reads:

$$\frac{\partial \omega}{\partial t} + u_j^b \frac{\partial \omega}{\partial x_j} = C_{1\omega} \frac{\omega}{k} R_{ij}^{bt} \frac{\partial u_i^b}{\partial x_j} + \frac{\partial}{\partial x_j} \left[ \left( \nu^b + \frac{\nu_t^b}{\sigma_\omega} \right) \frac{\partial \omega}{\partial x_j} \right] - C_{2\omega} \omega^2 - C_{3\omega} \frac{2K(1-t_{mf})\alpha\omega}{\rho^b} - C_{4\omega} S_{US} \frac{\omega}{k\beta} \nu_b^t \frac{\partial \alpha}{\partial x_j} \left( \frac{\rho^a}{\rho^b} - 1 \right) g_j. \quad (23)$$

The different coefficient values can be found in table 5. Similar as the two-phase $k - \varepsilon$ turbulence model, the coefficients associated with the present two-equation closure are adopted from their clear fluid counterpart. According to the numerical

experiments described in section 4, the coefficient $C_{3\omega}$ in the $\omega$ equation (23) is chosen to be 0.35. Again, the coefficient





associated with the buoyancy term $C_{4\omega} = 0$ is used in stably stratified condition, while it is set to 1 for unstably stratified condition. Due to different model configuration in $k - \varepsilon$ and $k - \omega$ models, the coefficient $B$ may be different. However, similar sensitivity of $B$ coefficient (see Eq. (19)) to the model results is observed, and it is left as the only free model calibration parameter.

**Table 5.** $k - \omega$ model coefficients.

| $C_\mu$ | $C_{1\omega}$ | $C_{2\omega}$ | $C_{3\omega}$ | $C_{4\omega}$ | $\sigma_k$ | $\sigma_\omega$ | $S_{US}$ |
|---|---|---|---|---|---|---|---|
| 0.09 | 5/9 | 3/40 | 0.35 | 0 or 1 | 2.0 | 2.0 | 1.0 |

Note that table 4 and 5 present the default coefficient values used in our implementation, which has been validated extensively with measured sediment transport data. However, they can be changed by setting the keyword and values in the files *constant/RASProperties* and *constant/twophaseRASProperties* (see appendix 1)

### 2.3 Particle phase stress

In sediment transport applications, the particle stress are important mechanisms to support particle immersed weight in concen-

trated regions of sediment transport (Hsu et al., 2004; Cheng, 2016). In these regions, the momentum exchanges due to particle collisions and/or enduring contacts exert dispersive stresses on a collection of particles. The particle phase stress tensor can be split into the normal and off-diagonal components that correspond to the particle pressure $\tilde{p}^a$ and the particle shear stress $\tilde{\tau}^a_{ij}$. Though, details varies with the model selection, the particle normal stresses (or pressure) can be generally classified into two contributions: a shear induced or collisional component (super-script 'a') and a permanent contact component (super-script

'ff') (Johnson and Jackson, 1987):.

$$\tilde{p}^a = p^{ff} + p^a, \tag{24}$$

The first term the particle pressure is due to enduring contact in highly concentrated region, where sediment bed is quasi-static/immobile. This normal pressure increases rapidly when the sediment concentration is close to maximum packing limit, and prevents unphysical sediment concentration in the sediment bed. Thus this element is important to model a full transport profile including the quasi-static sediment bed. The permanent contact component $p^{ff}$ is calculated as:

$$p^{ff} = \begin{cases} 0, \alpha < \alpha_{min}^{Fric} \\ Fr \dfrac{(\alpha - \alpha_{min}^{Fric})^{\eta_0}}{(\alpha_{max} - \alpha)^{\eta_1}}, \alpha \geq \alpha_{min}^{Fric}, \end{cases} \tag{25}$$

where $\alpha_{min}^{Fric} = 0.57$, $\alpha_{max} = 0.635$ for spheres and $Fr$, $\eta_0$ and $\eta_1$ are empirical coefficients. Following Cheng et al. (2017) the values are set to: $Fr = 0.05, \eta_0 = 3$ and $\eta_1 = 5$. Again, these coefficients can be reset in file *constant/kineticTheoryProperties* or *constant/granularRheologyProperties* by specifying the keywords and values (see appendix 1).





In modern sediment transport modeling framework, two major threads of modeling approach for shear-induced/collisional particle normal stress and shear stress are kinetic theory of granular flows and dense granular flow rheology. They are implemented in this version of SedFoam-2.0 (see subsection 2.3.1 and 2.3.2), respectively. As a result of different closures for particle normal stresses $p^a$, the closure for the total particle shear stress $\tilde{\tau}_{ij}^a$ also varies, and they are described in the next two

subsections.

### 2.3.1 Kinetic theory of granular flows

In the kinetic theory model, intergranular interactions are assumed to be dominated by binary collisions for low to moderate sediment concentration, and the collisional shear stresses are are quantified by particle velocity fluctuations represented by the granular temperature $\Theta$. The model is originally developed for dry granular flow consists of smooth, slightly inelastic,

spherical particles (Jenkins and Savage, 1983; Lun and Savage, 1987; Lun, 1991). Here, we adopt the model suggested by Ding and Gidaspow (1990), which takes into account the fluid phase. The balance equation for granular temperature is written as:

$$\frac{3}{2}\left[\frac{\partial \alpha \rho^a \Theta}{\partial t} + \frac{\partial \alpha \rho^a u_j^a \Theta}{\partial x_j}\right] = \left(-p^a \delta_{ij} + \tau_{ij}^a\right)\frac{\partial u_i^a}{\partial x_j} - \frac{\partial q_j}{\partial x_j} - \gamma + J_{int}, \tag{26}$$

where the first term on the RHS is the production of granular temperature, $q_j$ is the flux of granular temperature, $\gamma$ is the energy dissipation rate due to inelastic collision and $J_{int}$ is the production (or dissipation) due to the interaction with the carrier fluid

phase.

In the 1980s, dense phase kinetic theory of gases (Chapman and Cowling, 1970) was applied to granular flow by many researchers (Chepurniy, 1984; Jenkins and Savage, 1983; Savage, 1988). Here, we adopt the closure of particle pressure proposed by Ding and Gidaspow (1990):

$$p^a = \rho^a \alpha [1 + 2(1+e)\alpha g_{s0}]\Theta, \tag{27}$$

where $e$ is the coefficient of restitution during the collision. With the binary collision assumption adopted in the kinetic theory

of granular flow, the radial distribution function $g_{s0}$ is introduced to describe the crowdiness of particle. In this study, we use the radial distribution function for dense rigid spherical particles gases of Carnahan and Starling (1969):

$$g_{s0} = \frac{2-\alpha}{2(1-\alpha)^3}. \tag{28}$$

Following Gidaspow (1994), the particle collisional stress is calculated as:

$$\tau_{ij}^a = 2\mu^a \, S_{ij}^a + \lambda\frac{\partial u_k^a}{\partial x_k}\delta_{ij}. \tag{29}$$

where $S_{ij}^a$ is the deviatoric part of sediment phase strain rate tensor:

$$S_{ij}^a = \frac{1}{2}\left(\frac{\partial u_i^a}{\partial x_j} + \frac{\partial u_j^a}{\partial x_i}\right) - \frac{1}{3}\frac{\partial u_k^a}{\partial x_k}\delta_{ij}, \tag{30}$$



Through the kinetic theory, the particle shear viscosity is calculated as a function of granular temperature and radial distribution function:

$$\mu^a = \rho^a d\sqrt{\Theta}\Big[\frac{4}{5}\frac{\alpha^2 g_{s0}(1+e)}{\sqrt{\pi}} + \frac{\sqrt{\pi}g_{s0}(1+e)(3e-1)\alpha^2}{15(3-e)} + \frac{\sqrt{\pi}\alpha}{6(3-e)}\Big]. \tag{31}$$

Similarly, the bulk viscosity is calculated as:

$$\lambda = \frac{4}{3}\alpha^2 \rho^a d g_{s0}(1+e)\sqrt{\frac{\Theta}{\pi}}. \tag{32}$$

The closure of granular temperature flux is assumed to be analogous to the Fourier's law of conduction:

$$q_j = -D_\Theta \frac{\partial \Theta}{\partial x_j}, \tag{33}$$

where the $D_\Theta$ is the conductivity of granular temperature, calculated as

$$D_\Theta = \rho^a d\sqrt{\Theta}\Big[\frac{2\alpha^2 g_{s0}(1+e)}{\sqrt{\pi}} + \frac{9\sqrt{\pi}g_{s0}(1+e)^2(2e-1)\alpha^2}{2(49-33e)} + \frac{5\sqrt{\pi}\alpha}{2(49-33e)}\Big]. \tag{34}$$

The dissipation rate due to inelastic collision is calculated based on that proposed by Ding and Gidaspow (1990):

$$\gamma = 3(1-e^2)\alpha^2 \rho^a g_{s0}\Theta\Big[\frac{4}{d}\Big(\frac{\Theta}{\pi}\Big)^{1/2} - \frac{\partial u_j^a}{\partial x_j}\Big]. \tag{35}$$

Due to the presence of carrier fluid phase, carrier flow turbulence can also induce particle fluctuations. Following Hsu et al. (2004), the fluid-particle interaction term can be expressed as:

$$J_{int} = \alpha K(2t_{mf}k - 3\Theta). \tag{36}$$

In order to extend the model capability to resolve the quasi-static/immobile sediment bed, the shear stress due to frictional
contact is modeled as:

$$\tau_{ij}^{ff} = 2\rho^a \nu_{Fr}^a S_{ij}^a. \tag{37}$$

where $\nu_{Fr}^a$ is the frictional viscosity. By following Srivastava and Sundaresan (2003), which combined the frictional normal stress from Johnson and Jackson's model (Eq. 25) and the frictional viscosity from Schaeffer (1987) model, and the friction viscosity is calculated by:

$$\nu_{Fr}^a = \frac{p^{ff}\sin(\theta_f)}{\rho^a \left(\| \mathsf{S}^a \|^2 + D_{small}^2\right)^{1/2}}, \tag{38}$$

where a constant friction angle $\theta_f$ is used. This frictional shear viscosity model has the capability to capture the transition
from solid-like behaviour to fluid-like behaviour of the sediment bed. A small number of $D_{small} = 10^{-10}s^{-1}$ is added in the denominator to ensure numerical stability when shear rate in the sediment bed becomes zero. In sediment transport, the



frictional component of particle pressure and particle shear stress play a definite role to ensure the existence of an immobile sediment bed and a low mobility layer of enduring contact can be modeled (Hsu et al., 2004).

The total shear stress $\tilde{\tau}_{ij}^a$ can be calculated as a sum of the collisional-kinetic component ($\tau_{ij}^a$) and a frictional component ($\tau_{ij}^{ff}$):

$$\tilde{\tau}_{ij}^a = \tau_{ij}^a + \tau_{ij}^{ff}. \tag{39}$$

All the simulations presented in this paper have been obtained using the closures summarized in table 6. Other models are available in OpenFOAM and it would be very easy to implement new ones.

**Table 6.** Kinetic theory closure models

| keywords | keywordValue | Formulation |
|---|---|---|
| *granularPressureModel* | *Lun* | Eq. (27) |
| *radialModel* | *CarnahanStarling* | Eq. (28) |
| *viscosityModel* | *Syamlal* | Eq. (31) |
| *conductivityModel* | *Syamlal* | Eq. (34) |
| *frictionalStressModel* | *SrivastavaSundaresan* | Eq. (38) |

### 2.3.2   Dense granular flow rheology

The other alternative for modeling the particle phase stress proposed in SedFoam-2.0 consists of the dense granular flow rheology or the so-called $\mu(I)$ rheology (GDRmidi, 2004; Forterre and Pouliquen, 2008). Such an approach has been used

with some success by Revil-Baudard and Chauchat (2013) and Chauchat (2017) to model turbulent sheet flows. Contrary to the kinetic theory of granular flows, the dense granular flow rheology is phenomenological, and it is based on dimensional analysis. Instead of separating the collisional shear stress and frictional shear stress, the total particle phase shear stress is related to total particle pressure $\tilde{p}^a$ by a dynamic friction coefficient $\mu$ (Jop et al., 2006):

$$\tilde{\tau}_{ij}^a = \mu(\mathrm{I})\, \tilde{p}^a\, \frac{S_{ij}^a}{\sqrt{2\, S_{ij}^a \cdot S_{ij}^a}}, \tag{40}$$

The dynamic friction coefficient $\mu$ depends on the dimensionless controlling number 'I'. Depending on the local Stokes and particulate Reynolds numbers, the regime of the granular flow rheology can change from free fall or grain inertia regime to viscous and turbulent regimes (Andreotti et al., 2013), and the definition of the controlling parameter 'I' are different accordingly. In SedFoam-2.0, the grain inertia and viscous regimes have been implemented and can be selected using the keywords 'MuI' and 'MuIv' respectively. To be consistent with the definitions used in the kinetic theory, we still introduce the

the particle shear viscosity $\mu^a$ and frictional shear viscosity $\nu_{Fr}^a$. However, we simply set $\mu^a = 0$, and the define the frictional





shear viscosity alone as (Chauchat and Médale, 2014):

$$\nu_{Fr}^a = \frac{\mu(I)\,\tilde{p}^a}{\rho^a\,(\parallel \mathsf{S}^\mathsf{a} \parallel^2 + D_{small}^2)^{1/2}},\tag{41}$$

where $\parallel \mathsf{S}^\mathsf{a} \parallel$ is the norm of the shear rate tensor (Eq. 30). $D_{small}$ is the regularization parameter, which is introduced to avoid singularity. It is set to be $D_{small} = 10^{-6}s^{-1}$ for all the simulations of granular rheology presented herein except stated otherwise.

#### 2.3.2.1 Viscous regime

In the viscous regime, the friction coefficient $\mu$ depends on the viscous number $I_v = \parallel \nabla \mathbf{u^a} \parallel \nu^b / (\rho^b\,\tilde{p}^a)$ and is calculated as:

$$\mu(I_v) = \mu_s + \frac{\mu_2 - \mu_s}{I_0/I_v + 1},\tag{42}$$

For neutrally buoyant beads, the typical values are: $\mu_s = 0.32$, $\mu_2 = 0.7$ and $I_0 = 0.005$ (Boyer et al., 2011). The viscous regime occurs when the Stokes number $St = d\sqrt{\rho^a\,p^a}/(\rho^b\nu^b)$ as defined by Cassar et al. (2005), is lower than unity.

Concerning the shear induced contribution to the particle pressure, Boyer et al. (2011) proposed the following empirical formula for the dependance of sediment concentration $\alpha$ on the viscous number $I_v$:

$$\alpha(I_v) = \frac{\alpha_{max}}{1 + B_\phi I_v^{1/2}}.\tag{43}$$

where $B_\phi = 1/3$ is a parameter of the dilatancy law (Maurin et al., 2016; Chauchat, 2017). Inverting equation (43) and substituting the definition of the inertial number $I_v$ gives the following expression for the shear induced pressure:

$$p^a = \left(\frac{B_\phi\,\alpha}{\alpha_{max} - \alpha}\right)^2 \nu^b \parallel \mathsf{S}^\mathsf{a} \parallel.\tag{44}$$

With equation (44), the total particle pressure $\tilde{p}^a$ can be calculated by Eq. (24). The frictional viscosity is defined by equation (41) with $\mu(I)$ substituted by the $\mu(I_v)$ as shown in Eq. (42).

#### 2.3.2.2 Grain inertia regime

In the grain inertia regime, the friction coefficient depends on the inertial number $I = \parallel \nabla \mathbf{u^a} \parallel d\sqrt{\rho^a/\tilde{p}^a}$ and is calculated as:

$$\mu(I) = \mu_s + \frac{\mu_2 - \mu_s}{I_0/I + 1},\tag{45}$$

with $d$ the particle diameter, $\mu_s$ the static friction coefficient, $\mu_2$ an empirical dynamical coefficient and $I_0$ an empirical constant of the rheology. For glass beads in air the typical values are: $\mu_s = 0.38$, $\mu_2 = 0.64$ and $I_0 = 0.3$ (Jop et al., 2006).

Concerning the shear induced contribution to the particle pressure, it can be obtained from the dilatancy law $\alpha(I)$ as proposed by Boyer et al. (2011) for the viscous regime of the granular flow rheology. The adaptation to the inertial regime leads to the expression suggested by Maurin et al. (2016):

$$\alpha(I) = \frac{\alpha_{max}}{1 + B_\phi I}.\tag{46}$$



Inverting equation (46) and substituting the definition of the inertial number $I$ gives the following expression for the shear induced pressure (Chauchat, 2017):

$$p^a = \left( \frac{B_\phi\, \alpha}{\alpha_{\max} - \alpha} \right)^2 \rho^a d \parallel \mathsf{S}^a \parallel^2 . \tag{47}$$

Similar to those in the viscous regime, the total particle pressure $\tilde{p}^a$ can be calculated by Eq. (24) and the frictional viscosity is defined by equation (41) with $\mu(I)$ substituted by the $\mu(I)$ as Eq. (45).

The rheology has been originally stated for steady uniform granular flows and this shear induced pressure term induces a very strong coupling between the wall normal and the stream-wise components of the particle phase momentum balance equation. The granular flow rheology has been used to simulate with success the transient flows such as the granular column collapse configuration Lagrée et al. (2011) but the simulations have been performed at fixed volume concentration meaning that the shear induced pressure term was neglected. In order to stabilize the model, a relaxation is added: $p^a_{new} = relaxPa\, p^a + (1 - relaxPa)\, p^a_{old}$ with a relaxation factor $relaxPa$ that can be modified from the file *constant/granularRheologyProperties*. This is equivalent to assuming a relaxation in time for the dilatancy effect, the granular material do not dilate instantaneously to an imposed shear rate, which is physically justified.

The different closure laws implemented in sedFoam for the dense granular flow rheology are summarized in table 7.

**Table 7.** Dense granular flow rheology models

| kewords | | *none* | *Coulomb* | *MuI* | *MuIv* |
|---|---|---|---|---|---|
| *FrictionModel* | $\mu =$ | 0 | $\mu_s$ | Eq. (45) | Eq. (42) |
| *PPressureModel* | $p^a =$ | 0 | 0 | Eq. (47) | Eq. (44) |

## 3 Numerical implementation

The numerical implementation of the present Eulerian two-phase flow sediment transport model is based on an open-source finite volume CFD library called OpenFOAM. Taking advantage of the numerical discretization schemes and framework of finite volume solvers in OpenFOAM, the two-phase flow governing equations are implemented by modifying the solver twoPhaseEulerFoam (Rusche, 2002; Weller, 2002; Peltola, 2009). OpenFOAM uses the Finite Volume Method (FVM) over a collocated grid arrangement. The collocated arrangement stores all dependent variables at the cell center and the same Control Volume (CV) is used for all variables to minimize the computational effort. The advantage of the FVM is that the system of partial differential equations can be discretized on arbitrary three-dimensional structured or unstructured meshes. Thus, complex geometries can be easily handled. The Gauss theorem is applied to the convection and diffusion terms leading to conservative schemes.





To illustrate the numerical discretization, the fluid phase momentum equation are taken as an example. Rearranging the fluid phase momentum equation (Eq. 4) by dividing $\beta\rho^b$, the resulting equation can be written as

$$\frac{\partial \mathbf{u}^b}{\partial t} + \nabla \cdot (\mathbf{u^b u^b}) - (\nabla \cdot \mathbf{u^b})\mathbf{u^b} = -\frac{1}{\rho^b}\nabla p - \frac{\alpha K}{\rho^b}(\mathbf{u^b} - \mathbf{u^a}) + \frac{K}{\rho^b}\frac{1}{\sigma_c}\nu^{bt}\nabla\alpha + \mathbf{g} + \frac{1}{\beta}\nabla \cdot \tau^\mathbf{b} \tag{48}$$

The last term in the above equation, the gradient of fluid phase shear stress, can be written as can be written according to equation (7) and expanded as follows:

$$\frac{1}{\beta}\nabla \cdot \tau^\mathbf{b} = \nabla \cdot \left(\nu_{Eff}^b \nabla \mathbf{u}^b\right) + \nu_{Eff}^b \frac{\nabla\beta}{\beta}\nabla \mathbf{u}^b + \frac{1}{\beta}\nabla \cdot \left\{\beta\,\nu_{Eff}^b\left[(\nabla \mathbf{u}^b + \nabla \mathbf{u}^{bT}) - \frac{2}{3}\nabla \cdot \mathbf{u}^b\right]\right\}. \tag{49}$$

In the above equation, the first two terms on the RHS are treated implicitly while the last two terms are treated explicitly. By substituting the expanded shear stress formulation in the momentum equation, the following equation is obtained:

$$\frac{\partial \mathbf{u}^b}{\partial t} + \nabla \cdot (\mathbf{u^b u^b}) - (\nabla \cdot \mathbf{u^b})\mathbf{u^b} \quad -\nabla \cdot \left(\nu_{Eff}^b \nabla \mathbf{u}^b\right) - \nu_{Eff}^b \frac{\nabla\beta}{\beta}\nabla \mathbf{u}^b + \frac{\alpha K}{\rho^b}\mathbf{u^b} = -\frac{1}{\rho^b}\nabla p$$

$$+ \frac{\alpha K}{\rho^b}\mathbf{u^a} + \frac{1}{\sigma_c}\frac{K\nu^{bt}}{\rho^b}\nabla\alpha + \mathbf{g} + \frac{1}{\beta}\nabla \cdot \left\{\beta\,\nu_{Eff}^b\left[\nabla \mathbf{u}^{bT} - \frac{2}{3}\nabla \cdot \mathbf{u}^b\right]\right\} \tag{50}$$

It is more convenient to rewrite the above equation into a matrix form:

$$\left[A^\mathbf{b}\right] \cdot \mathbf{u^b} = \mathbf{H^b} + \mathbf{R^b} - \frac{1}{\rho^b}\nabla p \tag{51}$$

The matrix $\left[A^\mathbf{b}\right]$ is composed of the diagonal terms of the algebraic system associated with equation (50), whereas $\mathbf{H^b}$ includes the off-diagonal terms and the source terms. $\mathbf{R^b}$ is composed of the explicit drag term, the turbulent suspension term, the gravity term and the explicit diffusion terms:

$$\mathbf{R^b} = \frac{\alpha K}{\rho^b}\mathbf{u^a} + \mathbf{g} + \frac{1}{\beta}\nabla \cdot \left\{\beta\,\nu_{Eff}^b\left[\nabla \mathbf{u}^{bT} - \frac{2}{3}\nabla \cdot \mathbf{u}^b\right]\right\} + \frac{1}{\sigma_c}\frac{K\,\nu^{bt}}{\rho^b}\nabla\alpha \tag{52}$$

The same process can be carried out for the solid phase momentun equation (3) that leads to:

$$\frac{\partial \mathbf{u}^a}{\partial t} + \nabla \cdot (\mathbf{u^a u^a}) - (\nabla \cdot \mathbf{u^a})\mathbf{u^a} - \frac{1}{\tilde{\alpha}}\nabla \cdot (\nu_{Fr}^a \nabla \mathbf{u}^a) - \nabla \cdot \left(\nu_{Eff}^a \nabla \mathbf{u}^a\right) - \nu_{Eff}^a \frac{\nabla\alpha}{\tilde{\alpha}}\nabla \mathbf{u}^a + \frac{\beta K}{\rho^a}\mathbf{u^a} = -\frac{1}{\tilde{\alpha}\rho^a}\nabla p^{ff} - \frac{1}{\rho^a}\nabla p$$

$$+ \frac{\beta K}{\rho^a}\mathbf{u^b} - \frac{1}{\sigma_c}\frac{\beta K\,\nu^{bt}}{\tilde{\alpha}\rho^a}\nabla\alpha + \mathbf{g} - \frac{1}{\tilde{\alpha}\rho^a}\nabla p^a + \frac{1}{\tilde{\alpha}}\left\{\nabla \cdot \left[(\alpha\nu_{Eff}^a + \nu_{Fr}^a)\nabla \mathbf{u}^{a\,T}\right] + \nabla \left[\left(\lambda - \frac{2}{3}(\alpha\nu_{Eff}^a + \nu_{Fr}^a)\right)\nabla \cdot \mathbf{u}^a\right]\right\} \tag{53}$$

where $\alpha$ at the denominator is substituted by $\tilde{\alpha} = \alpha + \alpha_{Small}$ to avoid dividing by zero when the solid phase volume concentration vanishes. This partial differential system of equations can also be written as a matrix equation as follows:

$$\left[A^\mathbf{a}\right] \cdot \mathbf{u^a} = \mathbf{H^a} + \mathbf{R^a} - \frac{1}{\rho^a}\nabla p, \tag{54}$$



where the source term $\mathbf{R^a}$ contains the following terms:

$$\mathbf{R^a} = \frac{\beta K}{\rho^a}\mathbf{u^b} - \frac{1}{\sigma_c}\frac{\beta K\ \nu^{bt}}{\tilde{\alpha}\rho^a}\nabla\alpha + \mathbf{g} - \frac{1}{\tilde{\alpha}\rho^a}\nabla p^a + \frac{1}{\tilde{\alpha}}\left\{\nabla\cdot\left[(\alpha\nu^a_{Eff} + \nu^a_{Fr})\nabla\mathbf{u}^{a\ T}\right] + \nabla\left[\left(\lambda - \frac{2}{3}(\alpha\nu^a_{Eff} + \nu^a_{Fr})\right)\nabla\cdot\mathbf{u}^a\right]\right\}$$

(55)

Following Rusche (2002) the terms involving the ratio of particle phase volume gradient to the volume concentration are treated at the cell face level in the predictor-corrector algorithm. However, we noted the exception of the particle phase normal

stress $p^{ff}$ gradient for which a reconstruction of the surface normal gradient at the cell centre allows to get more stable solutions.

The advantage of separating the right hand side of the momentum equations as the sum of two terms: $\mathbf{R}$ and $\mathbf{H}$, is for writing the pressure-velocity algorithm. A similar method as Rhie and Chow (1983) one can be applied for the gradient terms. The details of the velocity-pressure algorithm are presented in the next subsection.

## 3.1 Velocity-pressure algorithm

The pressure-velocity coupling and the consequent oscillations in the pressure fields are resolved using the Rhie and Chow method (Rhie and Chow, 1983). The PISO algorithm is used to solve fluid and particle velocities (Rusche, 2002; Weller, 2002; Peltola, 2009).

First, the intermediate velocities ($\mathbf{u^{a*}}$,$\mathbf{u^{b*}}$) are computed using the corresponding momentum equations (equationswithout

the pressure gradient term:

$$\mathbf{u^{a*}} = \left[A^a\right]^{-1}\mathbf{H^a},$$
$$\mathbf{u^{b*}} = \left[A^b\right]^{-1}\mathbf{H^b},$$

(56)

where $\left[A^a\right]^{-1}$ and $\left[A^b\right]^{-1}$ represent the inverse matrices of $\left[A^a\right]$ and $\left[A^b\right]$, respectively. These intermediate velocities do not satisfy the mass conservation equations (2) and (1). To enforce mass conservation for each phase, the pressure equation is constructed by considering the continuity equations for the mixture obtained as the summation of equations (2) and (1):

$$\int_{Vp}\nabla\cdot\left[\alpha\mathbf{u^a} + \beta\mathbf{u^b}\right]dV = \oint_S\left[\alpha_f\mathbf{u^a}|_f + \beta_f\mathbf{u^b}|_f\right]\cdot\mathbf{n}\ dS = 0$$

(57)

where the subscript $f$ denotes variables interpolated at the cell faces. The two expressions shown above are equivalent by using the Gauss theorem. At the discrete level, this equation is written as:

$$\sum_f\left[\alpha_f\Phi^a_f + \beta_f\Phi^b_f\right] = 0,$$

(58)

where $\Phi^a_f = \mathbf{u^a}|_f.\mathbf{n}|_fS_f$ and $\Phi^b_f = \mathbf{u^b}|_f.\mathbf{n}|_fS_f$ denote the fluid and particle phases velocity fluxes at the cell faces, respec-

tively, and $S_f$ is the cell face area associated with face $f$. In the present model, the method of Rhie and Chow (1983) is adopted





in order to avoid velocity-pressure decoupling and oscillations. The velocity correction equation is written as:

$$\mathbf{u^a} = \mathbf{u}^{a*} + \frac{\mathbf{R^a}}{[A^a]} - \frac{\nabla p}{\rho^a[A^a]} \quad \text{at the cell centre} \quad \text{or} \quad \Phi_f^a = \Phi_f^{a*} + \frac{\Phi_f^{Ra}}{[A^a]_f} - \frac{\nabla^\perp p|_f}{\rho^a[A^a]_f} \quad \text{at the cell faces}$$

$$\mathbf{u^b} = \mathbf{u}^{b*} + \frac{\mathbf{R^b}}{[A^b]} - \frac{\nabla p}{\rho^b[A^b]} \quad \text{at the cell centre} \quad \text{or} \quad \Phi_f^b = \Phi_f^{b*} + \frac{\Phi_f^{Rb}}{[A^b]_f} - \frac{\nabla^\perp p|_f}{\rho^b[A^b]_f} \quad \text{at the cell faces} \quad (59)$$

In order to simplify the notations, the intermediate velocity and face fluxes are denoted as $\tilde{\mathbf{u}}^{a/b}$ and $\tilde{\Phi}_f^{a/b}$, respectively, corresponding to the first two terms on the RHS of equations (59). The volume averaged velocity, $\mathbf{U}^*$, and the corresponding averaged flux, $\Phi_f^*$, at the predictor step are defined as:

$$\mathbf{U}^* = \alpha \tilde{\mathbf{u}}^a + \beta \tilde{\mathbf{u}}^{b*} \quad \text{or} \quad \Phi^* = \alpha_f \tilde{\Phi}_f^a + \beta_f \tilde{\Phi}_f^b$$

Taking the divergence of the volume averaged mixture velocity given by the velocity correction equation (59) and imposing the incompressibility constraint, $\nabla \cdot \mathbf{U} = \nabla \cdot (\alpha \mathbf{u^a} + \beta \mathbf{u^b}) = 0$, one can built the pressure equation as a function of the predicted velocity or predicted face fluxes:

$$\int_{Vp} \nabla \cdot \left[ \left( \frac{\alpha}{\rho^a[A^a]} + \frac{\beta}{\rho^b[A^b]} \right) \nabla p \right] dV = \int_{Vp} \nabla \cdot \mathbf{U}^* dV \quad \text{or} \quad \oint_S \left( \frac{\alpha_f}{\rho^a[A^a]_f} + \frac{\beta_f}{\rho^b[A^b]_f} \right) \nabla^\perp p|_f \, \mathbf{n}|_f \, dS = \oint_S \mathbf{U}^*|_f \, \mathbf{n}|_f \, dS.$$

$$(60)$$

The poisson equation for the pressure is then written in discretized form at the cell face level as:

$$\sum_f \left( \frac{\alpha_f}{\rho^a[A^a]_f} + \frac{\beta_f}{\rho^b[A^b]_f} \right) \nabla^\perp p|_f \, \mathbf{n}|_f \, S_f = \sum_f \Phi_f^*. \quad (61)$$

This equation leads to a matrix system written at the cell faces. The resulting algebraic system is usually solved using an multigrid solver (GAMG). The resulting pressure field $p^*$ is used for the correction step in which the fluid and particle phase velocities and face fluxes are corrected using equation (59):

$$\Phi_f^{a/b**} = \tilde{\Phi}_f^{a/b} - \frac{\nabla^\perp p^*|_f}{\rho^{a/b}[A^{a/b}]_f} \quad \text{and} \quad \mathbf{u}^{a/b**} = \tilde{\mathbf{u}}^{a/b} - \frac{\nabla p^*}{\rho^{a/b}[A^{a/b}]}, \quad (62)$$

The volume averaged flux is also corrected according to:

$$\Phi_f^{**} = \Phi_f^* - \left( \frac{\alpha_f}{\rho^a[A^a]_f} + \frac{\beta_f}{\rho^b[A^b]_f} \right) \nabla^\perp p^*|_f \, \mathbf{n}|_f \, S_f \quad (63)$$

In order to ensure the mass conservation an iterative procedure of N cycles is sometimes required. From our experience, three iterations (N=3) is usually enough for a convergence. The finite volume discretisation of the equations have not been shown here but all the details can be found in Jasak (1996) and Rusche (2002).





## 3.2 Summary of the solution procedure

The numerical solution procedure for the proposed two-phase flow model is outlined as follow:

1. Solve for sediment concentration $\alpha$, i.e., Eq. (1);

2. Update the volume concentration of fluid: $\beta = 1 - \alpha$;

3. Update the drag parameter $K$ in the drag term, e.g., Eqn (5);

4. Solve for the fluid turbulence closure, update $k$, $\varepsilon$ or $\omega$ (depends on the turbulence closure $k - \varepsilon$ or $k - \omega$), and then calculate the eddy viscosity and effective fluid total viscosity;

5. Solve for the particle phase stress (kinetic theory model or the dense granular rheology);

6. PISO-loop, solving velocity-pressure coupling for $N$ loops:

(a) Construct the coefficient matrix $\left[\mathrm{A^a}\right]$ and $\left[\mathrm{A^b}\right]$ and explicit array $\mathbf{H^a}$ and $\mathbf{H^b}$ using Eqn (54) and (51).

(b) update the other explicit source terms $\mathbf{R^a}$ and $\mathbf{R^b}$, Eq. (55) and (52).

(c) Calculate $\mathbf{u^{a*}}$, $\mathbf{u^{b*}}$ using equations (56) without fluid pressure gradient term;

(d) Construct and solve the pressure Eq. (61);

(e) Correct fluid and particle velocities after solving pressure and update fluxes Eqns (62)-(63);

(f) go to (a-e) if the number of loops is smaller than $N$ (no tolerance criteria).

7. Advance to the next time step

In the above solution procedure, the velocity-pressure coupling steps are looped for $N$ times. The advantage of this loop is to avoid velocity-pressure decoupling caused by the direct solving method. From our numerical practices, the loop number $N = 1$ to 3 is usually enough to give reasonably accurate results, and shows good convergence, especially for steady flows.

## 4   Model verification and benchmarking

In this section, two benchmarking cases are presented to validate/verify the numerical implementation of the model. The first one concerns the pure sedimentation of a suspension of non-cohesive spherical particles for which experimental data are available and the second one concerns the laminar bed-load problem for which an analytical solution exists.

### 4.1   Pure sedimentation

The first test case corresponds to a pure sedimentation of non-cohesive particles, the experimental dataset from Pham Van Bang et al. (2008) is used for this validation. The suspension consists of mono-dispersed spherical polystyrene beads of diameter





$d = 0.29 \pm 0.03$ mm, and of density $\rho^a = 1050$ kg/m³ in Rhodorsil silicone oil of viscosity $\nu^b = 2.01 \times 10^{-5}\ m^2.s^{-1}$ and of density $\rho^b = 950$ kg/m³. The suspension is initially well-mixed in a cylindrical container (base diameter 50 mm; height 100 mm) with an initial solid volume concentration ($\alpha^0 = 0.5$). The averaged concentration profiles are measured using a Proton MRI device (Ecole de Ponts ParisTech, Champs-sur-Marne, France) with a vertical resolution of about 1 mm and a temporal

frequency of 0.16 Hz (see Pham Van Bang et al. (2008) for details). This test case has been used by Chauchat et al. (2013) to validate a 1D vertical two-phase flow model. A script for computing the solution is provided in the tutorial folder coming with the release 2.0 of SedFoam-2.0.

The mesh is composed of 200 cells in the vertical direction with a uniform distribution over a height $h_0 = 0.06$ meters and the time step is set to $\Delta t = 10^{-3}\ s$. The numerical schemes for temporal and spatial derivatives are listed in table 8, and details

of these numerical schemes can be found in the user guide of OpenFOAM. The lateral boundaries are set to cyclic while the front and back boundaries are set to empty (*i.e.* 2D problem). At the top boundary, the pressure is fixed with a zero value and a zero gradient (i.e Neumann boundary conditions) is imposed on all the other quantities. At the bottom, the velocity of both phases are set to zero while a fixedFluxPressure condition is imposed for the pressure. The granular rheology is turned on and an effective fluid viscosity model from Boyer et al. (2011) is used.

**Table 8.** Numerical schemes in validation test case 4.1[a]

| Description | keywords | keywordValue | Formulation |
|---|---|---|---|
| time derivative | *ddtSchemes* | *Euler implicit* | $\partial \xi / \partial t$ |
| spatial gradient operation | *gradSchemes* | *Gauss linear* | $\nabla \xi$ |
| divergence operators | *divSchemes* | *Gauss limitedLinear* | $\nabla \cdot \xi$ |
| laplacian operators | *laplacianSchemes* | *Gauss linear corrected* | $\nabla \cdot (\nabla \xi)$ |

[a] $\xi$ denotes a dummy variable for the illustration purpose.

Figure 1 shows the comparison of the numerical results with the experimental data in terms of settling curves (a) and sediment concentration profiles (b). The settling curves shows the time evolution of the vertical position of the upper $Y_i^{up}$ and lower $Y_i^{low}$ sediment interfaces. The upper one represents the transition between the suspension at the initial volume concentration and the clear water above it while the lower one denotes the interface between the suspension and the granular bed at maximum packing fraction. They are defined as follows: $Y_i^{up} = \max\{y \mid \alpha \geq 0.5\alpha^0\}$ and $Y_i^{low} = \max\{y \mid \alpha \geq 0.5\ (\alpha_{max} + \alpha^0)\}$. The

agreement between the numerical simulation results and the experiments is very good, suggesting that the closures for the drag and the particle pressure allows to reproduce a pure sedimentation problem. This is the most basic test case for a two-phase flow sediment transport model.





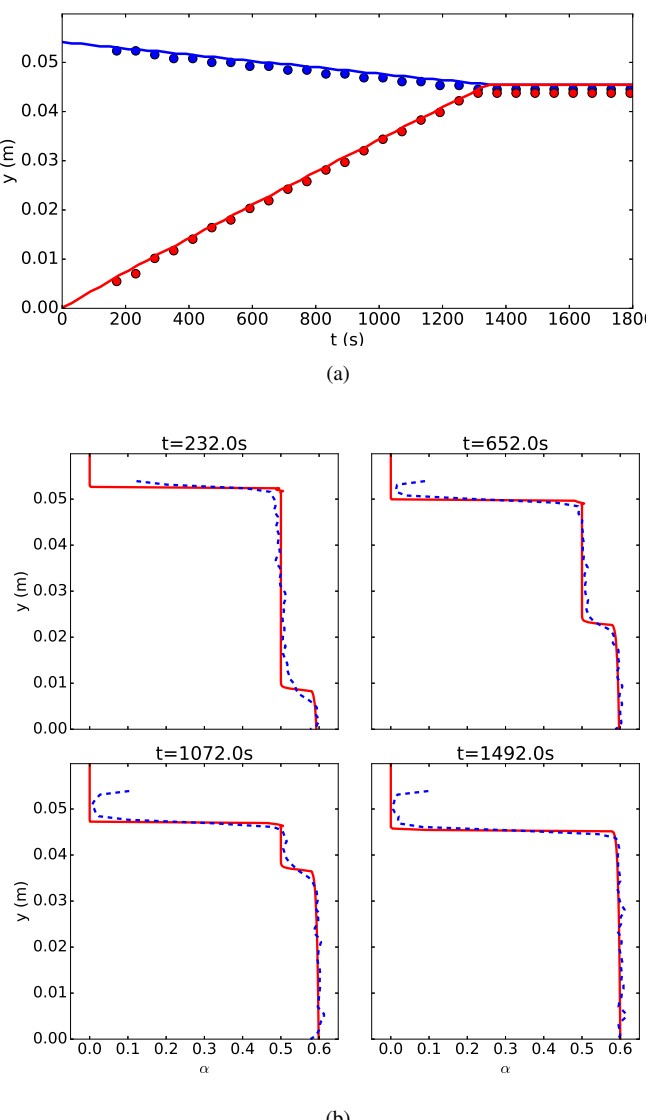

**Figure 1.** Comparison of two-phase flow model results with experiments of Pham Van Bang et al. (2008) a) Settling curves: time evolution of the lower and upper interface positions (circles: experiments ; lines: model) and b) Profiles of sediment concentration (dashed blue lines: experiment ; solid red lines: model).





## 4.2 Laminar bed-load

The second test case is inspired by Chauchat and Médale (2010) in which an analytical solution for laminar bed-load driven by a Poiseuille flow has been used to verify a three-dimensional numerical model. The solution is based on a Coulomb rheology for the solid phase and the Einstein's mixture viscosity of the fluid phase. The details concerning the analytical solution can

be found in Ouriemi et al. (2009) and will not be further detailed herein. A script for computing the solution is provided in the tutorial folder coming with the release of SedFoam-2.0.

The numerical domain setup is based on Aussillous et al. (2013) experimental configuration. The channel height is $h_0 = 0.065$ m, the particles are made of PMMA with a density $\rho^a = 1190$ kg/m$^3$ and a diameter $d = 2 \times 10^{-3}$ m. The fluid density is $\rho^b = 1070$ kg/m$^3$ and the kinematic viscosity is $\nu^b = 2.52 \times 10^{-4}$ m$^2$/s. The pressure gradient is fixed to gradPMEAN=100

kg.m$^{-2}$.s$^{-2}$. The vertical domain is discretized into 200 uniform cells, and the time step is $\Delta t = 10^{-3}$ s. The numerical schemes are identical to the pure sedimentation case in section 4.1 (see table 8). The lateral boundaries are set to cyclic while the front and back boundaries are set to empty (*i.e.* 2D problem). The velocity of both phases are set to zero at the top and bottom boundaries while the pressure is fixed to a zero at the top boundary and a fixedFluxPressure condition is imposed at the bottom boundary.

The granular rheology is a Coulomb rheology (see table 7) and an Einstein mixture fluid viscosity model (see table 2) is used. Figure 2.a shows the comparison of the numerical solution with the analytical solution from Ouriemi et al. (2009) in terms of sediment concentration (left panel), velocity (middle panel) and particle pressure (right panel) profiles. In the analytical solution, the sediment concentration profile is a step function with no particles in the upper half of the domain and with the maximum packing concentration in the lower half. The two-phase numerical model, based on continuous assumptions, is not

able to reproduce exactly this sharp sediment concentration transition. This is because that the sediment concentration profile is obtained using the momentum balance between the gravity and the permanent contact contribution to the particle pressure (Eq. 25). Despite this slight discrepancy, the numerical solution in terms of velocity profiles is in very good agreement with the analytical solution. Because the granular phase viscosity is directly related to the particle pressure (Eq. 40), the key issue for the granular rheology model is in the accurate prediction of the particle pressure profile. The comparison presented in the

right panel shows that even if the agreement in sediment concentration profile is not perfect, the particle pressure profile is very close to the analytic solution. This explains the very good numerical prediction of the velocity profile.

In Fig. 2.b, the exact same problem is solved using the non-linear dense granular flow rheology $\mu(I)$ (see table 7). This solution is compared with the solution of the 1D code described in Aussillous et al. (2013). The very good agreement between the two numerical solutions allows to validate the implementation of the $\mu(I)$ rheology in SedFoam-2.0. It should be pointed

out that, for this problem, the shear induced pressure contribution is not turned on explaining why the sediment concentration and the particle pressure profiles are not affected by the change in the granular rheology.





(a)

(b)

**Figure 2.** Comparison of the streamwise velocity profiles for the flow of a Newtonian fluid over a granular bed having a Coulomb rheology between two infinite parallel planes obtained by numerical simulations with the analytical solution of Ouriemi et al. (2009) in terms of sediment concentration (left panel), velocity profiles (middle panel) and particle pressure (right panel) profiles. In subfigure (b) the same flow conditions are used but the granular rheology is the $\mu(I)$ rheology and the results are compared with a one-dimensional numerical solution presented in Aussillous et al. (2013), the analytical solution of Ouriemi et al. (2009) is plotted for reference corresponding to the same pressure gradient. The panels are the same as in subfigure (a).





## 5  Applications

In this section, SedFoam-2.0 is applied to two sediment transport problems, the unidirectional turbulent sheet flow and the scour at an apron. For both cases, experimental data or empirical formula are used to validate the model. The first application is dedicated to demonstrate the model capability in the sheet flows, while the second application shows the multi-dimensional

feature of the present solver.

### 5.1  Turbulent sheet flows

This case is based on Revil-Baudard et al. (2015) experimental configuration for unidirectional sheet flows. The particles are non-spherical lightweight PMMA particles with density $\rho^a = 1190$ kg.m$^{-3}$ and diameter $d = 3 \pm 0.5$ mm. The measured settling velocity is $W_{fall} = 0.056$ m.s$^{-1}$. The fluid is water with density $\rho^b = 1000$ kg.m$^{-3}$ and kinematic viscosity $\nu^b = 10^{-6}$

m$^2$.s$^{-1}$. The water depth is $h_f = 0.17$ m, the energy slope is $S_f = 0.19$ % and the mean velocity is $\bar{U} = 0.52$ m.s$^{-1}$.

In the numerical configuration, the flow is driven by a pressure gradient (gradPMEAN=18.639 kg.m$^{-2}$.s$^{-2}$). The mesh is composed of 400 cells in the vertical direction with a uniform distribution and the time step is set to $\Delta t = 2\ 10^{-4}$ s. Again, the numerical schemes are identical to ones listed in table 8. The lateral boundaries are set to cyclic while the front and back boundaries are set to empty (*i.e.* 2D problem). At the top boundary, the pressure is fixed with a zero value and a zero gradient

is imposed on all the other quantities. At the bottom, the velocity of both phases are set to zero, a fixedFluxPressure condition is imposed for the pressure and a zeroGradient boundary condition is used for the TKE and the TKE dissipation ($\varepsilon$ and $\omega$).

The results are presented in Fig. 3 for velocity profiles (left panel), sediment concentration profile (center panel) and Reynolds shear stress profile (right panel). The numerical results are compared with the measurements reported in Revil-Baudard et al. (2015). Different combinations of granular stress model and turbulence model are presented. In subfigure (a) the

$\mu(I)$ rheology is used in combination with the Mixing Length (ML), $k-\omega$ and $k-\varepsilon$ models while in subfigure (b) the Kinetic Theory (KT) model is used in conjunction with the $k-\omega$ and $k-\varepsilon$ turbulence models. Note that the Mixing Length model has not been used with the kinetic theory as equation (36) requires an estimation of the TKE which is not straightforward to estimate when using a mixing length model. Amongst these different configurations, the $\mu(I)$ rheology coupled with the mixing length model (subfigure a) corresponds to the model proposed by Revil-Baudard and Chauchat (2013) and Chauchat (2017)

and the kinetic theory coupled with the $k-\varepsilon$ (subfigure b) corresponds to the model proposed by Hsu et al. (2004) and Cheng et al. (2017). The mixing length model has been calibrated based on Revil-Baudard et al. (2015) data using the $\mu(I)$ rheology (Chauchat, 2017) and the von Karman constant is reduced to $\kappa = 0.225$ to model a significant turbulence damping under sheet flow conditions. Also the rheological parameters of the $\mu(I)$ rheology are identical to the ones proposed by Chauchat (2017) and they are summarized in table 9.

In the $k$, $\varepsilon$ and $\omega$ equations, the turbulence damping is related to the drag term which is controlled by $t_{mf}$ that parameterize the correlation between particles and fluid fluctuating motions $t_{mf} = e^{-B\ St}$ ($t_{mf} = 1$ at low Stokes numbers and $t_{mf} = 0$ at high Stokes numbers). Based on Cheng et al. (submitted) LES simulations the parameter $B$ should be taken at 0.25 when using the kinetic theory. The same value has been obtained by calibrating this coefficient based on Revil-Baudard et al. (2015)





**Table 9.** Physical parameters for the numerical simulations of Sumer et al. (1996) experiments

| case | $S_f$ | $h_0$ | $z_{int}^0$ | $\theta$ | $W_{fall}/u_*$ | $S_{US}$ | $\mu_s$ | $\mu_2$ | $I_0$ | $B_\phi$ | $\alpha_{max}$ |
|------|-------|-------|-------------|----------|----------------|----------|---------|---------|-------|----------|----------------|
| | (-) | (m) | (m) | (-) | (-) | (-) | (-) | (-) | (-) | (-) | (-) |
| Revil-Baudard et al. | 0.0019 | 0.17 | 0.0211 | 0.44 | 1.0 | 2.2727 | 0.52 | 0.96 | 0.6 | 0.66 | 0.55 |
| SUM A | 0.0079 | 0.104 | 0.0526 | 1.38 | 1.04 | 2.2727 | 0.38 | 0.82 | 0.6 | 0.66 | 0.6 |
| SUM B | 0.0091 | 0.104 | 0.0521 | 1.63 | 0.95 | 2.2727 | 0.38 | 0.82 | 0.6 | 0.66 | 0.6 |
| SUM C | 0.0105 | 0.104 | 0.0516 | 2.18 | 0.84 | 2.2727 | 0.38 | 0.82 | 0.6 | 0.66 | 0.6 |

data using the kinetic theory of granular flows and the k-$\varepsilon$ model. The results are shown in Fig. 3b). For the $k-\omega$ model, the additional damping terms are similar, the $B$ value has been kept the same as for the $k-\varepsilon$. However, the $C_{3\omega}$ value has been tuned to recover almost the same velocity profile in the dense sheet flow layer as with the $k-\varepsilon$ model. A value of $C_{3\omega} = 0.35$ has been obtained which is very close to the value of $C_{3\omega} = 0.4$ reported by Amoudry (2014).

After this calibration of the turbulence models, the results obtained with the different combinations of granular stress and turbulence models are discussed. In terms of velocity profiles, both the $\mu(I)$ rheology and the kinetic theory, are able to reproduce reasonably well the measurements provided that the turbulence model is adequately tuned. However, when using the $k-\omega$ or $k-\varepsilon$ model with the $\mu(I)$ rheology, the velocity profiles are underestimated suggesting that the dissipation in the sheet flow layer is too strong and the velocity gradients are too small. The $B$ value could be re-calibrated to give a better result

but this is not the purpose of the present contribution. A general conclusion is that the numerical solution does not depend too much on the choice of the turbulence model in between the $k-\omega$ and $k-\varepsilon$ models. Concerning the sediment concentration profile, none of the two granular stress models is able to recover the measurements in the denser part of the sheet flow layer. According to Revil-Baudard et al. (2015), this discrepancy might be related to the very strong near bed intermittency and the assumptions used in the present turbulence-averaged formulation may be too simple. It may require a 3D turbulence-resolving

numerical simulation approach to capture this feature (Cheng et al., submitted).

     In order to further assess the model, the same combinations of granular stress and turbulence models are applied to Sumer et al. (1996) experimental configurations corresponding to Shields numbers in the range $\theta \in [1.38; 2.1]$. For this configuration the particles are made of acrylic, density $\rho^a = 1140$ kg.m$^{-3}$, and are of cylindrical shape with a mean diameter $d = 2.6$ mm. The measured settling velocity is $W_{fall} = 0.073$ m.s$^{-1}$. The fluid is water, density $\rho^b = 1000$ kg.m$^{-3}$ and kinematic viscosity

$\nu^b = 10^{-6}$ m$^2$.s$^{-1}$. The physical parameter for the three cases SUMA, SUMB, SUMC are summarized in table 9.

     The results in term of velocity, concentration and shear stress profiles are presented in Fig. 4. Using the $\mu(I)$ rheology with the mixing length model leads to an underestimation of the velocities as well as the velocity gradient in the sheet layer, at least for the first two cases. This could be explained by a smaller von Karman constant. The results obtained using the $\mu(I)$ rheology with the $k-\omega$ and $k-\varepsilon$ turbulence models again give very similar results that are in quite good agreement with

the measured profiles. When using the kinetic theory and the $k-\varepsilon$ model, the velocity profiles and the velocity gradients are overestimated in the sheet layer. Concerning the concentration profiles they are all very similar and the results are not very sensitive to the different combinations of granular stress and turbulence models. For the shear stress profiles, it is observed that




using the $\mu(I)$ rheology all the profiles are very close one to the other. However, when using the kinetic theory, the Reynolds shear stress penetrate deeper into the sheet layer than when using the $\mu(I)$ rheology. This is probably due to the presence of the fluid-particle interaction term in the granular temperature equation (Eq. 36).

Different hypotheses can be proposed to explain the discrepancies presented above. According to Maurin et al. (2016), the
$\mu(I)$ rehology is able to describe accurately the dense granular flow regime in turbulent sheet flows but it fails to predict the granular shear stress in the more dilute suspended layer at intermediate concentrations ($\alpha \leq 0.3$). It is therefore possible that the turbulence model is tuned in a way that the turbulent stress is overestimated leading to a underestimate the streamwise velocity. Concerning the kinetic theory, it is well known that it only applies when particle collisions are binary (Jenkins, 2006), *i.e.* when the concentration is not too high ($\alpha \leq 0.3$). For higher concentration, the particle-particle collisions involve more
than two particles and the particle-particle contacts become "chattering". Jenkins (2006) proposed an extended kinetic theory that can tackle this problem. Moreover, the theory proposed by Berzi (2011) and Berzi and Fraccarollo (2013) has been applied with success to study turbulent sheet flows. Furthermore, we believe that the frictional stress model used with the kinetic theory is too simple and can not reproduce the granular behavior in the denser part of the sheet layer whereas the $\mu(I)$ rheology is better suited. This is also supported by experimental observations from Capart and Fraccarollo (2011), the authors showed that
the thickness of the dense frictional layer increases with the Shields number. A way to improve the kinetic theory would be to incorporate the latest theoretical developments proposed by Jenkins (2006) as done by Berzi (2011) and Berzi and Fraccarollo (2013) but this is beyond the scope of the present contribution.

The focus of the present work is to demonstrate the capabilities included in SedFoam-2.0  to model unidirectional sheet flows. In particular, the comparison of different combinations of granular stress and turbulence models in the same numerical
framework is presented for the first time. This opens new perspectives on the formulation of a comprehensive two-phase flow model for sediment transport and it allows to discuss more objectively the advantages and drawbacks of each modeling choices.

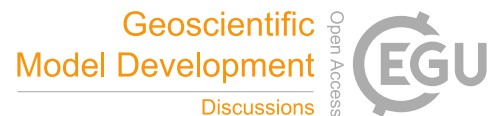



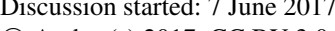

**Figure 3.** Comparison of two-phase numerical results with experiments of Revil-Baudard et al. (2015) in terms of velocity profiles (left panel), sediment concentration (center panel) and Reynolds shear stress (right panel) using the dense granular flow rheology ($\mu(I)$) with the three turbulence models (Mixing Length ML, $k-\omega$ and $k-\varepsilon$) in (a) and the Kinetic -Theory of granular flows (KT) with the three turbulence models (Mixing Length ML, $k-\omega$ and $k-\varepsilon$) in (b).







**Figure 4.** Comparison of two-phase numerical results with experiments of Sumer et al. (1996) in terms of velocity profiles (left panel), sediment concentration (center panel) and Reynolds shear stress (right panel) using the dense granular flow rheology ($\mu(I)$) with the three turbulence models (Mixing Length ML, $k - \omega$ and $k - \varepsilon$) and the kinetic theory of granular flows with the $k - \varepsilon$ turbulence model.



## 5.2  Scour at an apron

In order to demonstrate the multi-dimensional capability of SedFoam-2.0, the fourth test case corresponding to the development of the scour downstream an apron. Following the numerical study of Amoudry et al. (2008) and Cheng et al. (2017), the problem is simplified as presented in Fig. 5.

The sediment bed is made of sand, density $\rho^a$ = 2650 kg.m$^{-3}$ and diameter d= 0.25 x 10$^{-3}$m. The fluid is water with density $\rho^b$= 1000 kg.m$^{-3}$ and kinematic viscosity $\nu^b$ = 10$^{-6}$ m$^2$.s$^{-1}$. The flow depth is fixed to $h_0$ = 0.15 m, and the initial bed depth is $h_b$ = 0.05 m. The length of the domain downstream of the apron is $L_x$ = 1 m. A uniform grid is used in the streamwise direction ($\Delta x = 10^{-3}$ m) while the mesh is refined vertically at the bed interface ($\Delta y \in [1.15 \times 10^{-4} \ m \ - 1.15 \times 10^{-2} \ m]$ in the water column and $\Delta y \in [1.15 \times 10^{-4} \ m \ - 4.66 \times 10^{-4} \ m]$ in the sediment bed).

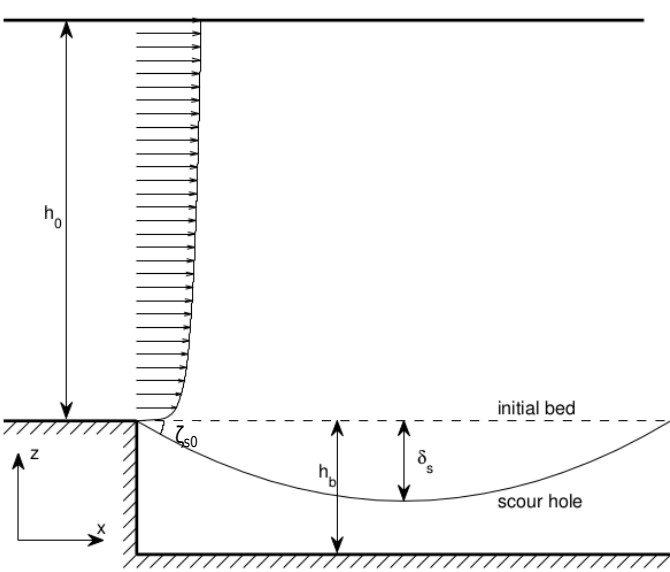

**Figure 5.** Sketch of the scour downstream an apron (source: Cheng et al. (2017)).

The bottom boundary, the lower part of the inlet (forming the step) and of the outlet are set as wall boundaries. The upper part of the inlet is an inlet boundary where the velocity profile is imposed according to the rough wall log law (eq. 64) and turbulent quantities are imposed as constant values following recommendation from esi group[1]. At the outlet, a directionMixed boundary condition is used for the velocities of both phases (zeroGradient for streamwise component and zero value for the vertical component) and the hydrostatic pressure is imposed. The top boundary is set as a symmetry plane. In all the simulations

in this case, the numerical schemes are also similar to the ones in table 8, except that 'Gauss limitedLinear 1' is used for the divergence operators. As initial condition, the velocity of both phases, the sediment concentration, the TKE and the TKE

---

[1]https://myesi.esi-group.com/tipstricks/guidelines-specification-turbulence-inflow-boundaries, requires a login





dissipation variables ($\varepsilon$ or $\omega$) are set based on one-dimensional simulation results using funkySetFields. The details of the boundary conditions are summarized in table 10. The rough wall log-law is written as

$$\frac{u}{u_*} = \frac{1}{\kappa} ln \left( \frac{30y}{k_s} \right),$$
(64)

where $u_* = 3.69$ cm.s$^{-1}$ is the bed friction velocity, $\kappa$=0.41 is the von Karman constant, and $k_s$=2.5d is the Nikuradse rough-
ness length.

**Table 10.** Summary of the boundary conditions implemented in the 2D scour downstream of an apron configuration: zG = *zeroGradient*, fV = *fixedValue*, dM = *directionMixed*, fFP = *fixedFluxPressure* and hp=hydrostatic pressure.

| Boundary | type | $\alpha$ | k | $\varepsilon$ or $\omega$ | $u^a$ | $u^b$ | p | $\Theta$ (for Kinetic Theory) |
|---|---|---|---|---|---|---|---|---|
| top | patch | zG | zG | zG | zG | zG | zG | zG |
| bottom | wall | zG | zG | zG | fV, $u^a$=0 | fV, $u^b$=0 | fFP | zG |
| inlet (flow) | patch | 1D profile | fV, $k = 1 \times 10^{-4}$ | zG | 1D profile | 1D profile | zG | fV, $k = 1 \times 10^{-6}$ |
| inlet (sed) | wall | zG | fV, $k = 1 \times 10^{-12}$ | zG | fV, $u^a$=0 | fV, $u^b$=0 | zG | zG |
| outlet (flow) | patch | zG | zG | zG | dM | dM | hp | zG |
| outlet (sed) | wall | zG | fV, $k = 1 \times 10^{-12}$ | zG | fV, $u^a$=0 | fV, $u^b$=0 | hp | zG |

Four combinations of fluid turbulence models ($k - \varepsilon$ and $k - \omega$) and granular stress models (KT and $\mu(I)$)) have been used (see table 11 for reference). Fig. 6 shows three snapshots of sediment concentration contour at different instants during the scour process, using $k - \varepsilon$ and kinetic theory (left pannels) and $k - \omega$ and $\mu(I)$ granular rheology (right pannels). At t = 10 s, the development of a scour hole near the inlet can be identified (see Fig. 6, top panels). As time elapses, the maximum scour
depth increases and the scour perturbation is propagating downward. The snapshots presented in Fig. 6 show that results are qualitatively in agreement one with the other one using the different closures. According to experimental studies (e.g. Breusers, 1967; Breusers and Raudkivi, 1991), the development of the scour hole is rapid at the initial stage, and eventually reaches an equilibrium state. Breusers (1967) has suggested an empirical law to describe the rapid initial development of the scour hole:

$$\frac{\delta_s}{h_0} = \left( \frac{t}{T_s} \right)^{n_s}$$
(65)

where $T_s$ is a characteristic timescale, and the exponent $n_s$ characterizes the speed of the scour development. Notice that eq. (65) only describes the initial development of the scour depth, and the equilibrium scour depth can not be determined from this empirical formula. As the scour depth increases, the flow velocity reduces near the sediment bed, when the flow becomes weak enough, thus it is below the criteria for sediment motion, an equilibrium scour depth can be obtained. The equilibrium scour shape is generally independent of the flow velocity and grain size if the Shields parameter is sufficiently large compared
with the critical Shields parameter (Laursen, 1952; Chane, 1984). The development of the upstream bed angle $\zeta_{s0}$ can reach an equilibrium more rapidly. Breusers (1967) proposed the following empirical formula:

$$\frac{\zeta_{s0}}{\zeta_{s0}^\infty} = \left( 1 - e^{-t/T_{\zeta_{s0}}} \right)$$
(66)





**Figure 6.** Sediment concentration contour at different time during the scour process using k-$\varepsilon$ and kinetic theory (left pannels) and k-$\omega$ and $\mu(I)$ granular rheology.



where $T_{\zeta s0}$ is the equilibrium timescale for upstream bed angle and $\zeta_{s0}^\infty$ is the upstream bed angle at equilibrium.

In Fig. 7, the numerical results for the four simulations presented above are shown in terms of these two quantities together with a best fit of the two empirical formula Eqs (65) and (66). A summary of the fitted parameters is given in table 11. First, the model is able to reproduce the power law for the initial development of the scour depth with values of $n_s$ in the range reported 5 by other studies with higher $T_s$ values. The fitted values for the upstream bed angle are also in agreement with former studies.

In conclusion, this test case shows good capability of the proposed two-phase flow model to deal with multi-dimensional flow configurations. Further work is needed to improve the model validation as well as the model sensitivity to flow turbulence and rheological parameters. This requires more detailed experimental data that, to the best of our knowledge, are not available at present.

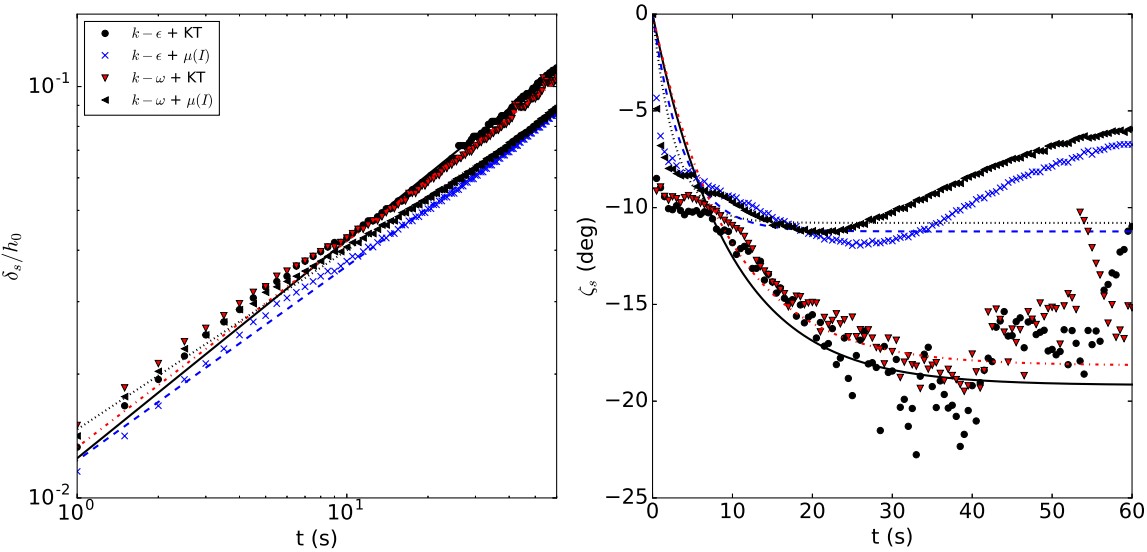

**Figure 7.** Numerical results for temporal evolution of the normalized maximum scour depth and upstream bed angle. Different lines represent the best-fit curves for each run for which the parameters are given in table 11.

## 6    Conclusions

In this paper, a comprehensive two-phase flow model for sediment transport applications has been presented and the details concerning its implementation in openFOAM has been given. The proposed model provide different options for the modeling of flow turbulence (mixing length, $k-\varepsilon$ or $k-\omega$) and inter-granular stress (kinetic theory of granular flows or dense granular flow rheology). The first validation test case presented on sedimentation of mono-disperse spherical suspension allow to validate the 15 numerical implementation of the pressure velocity coupling as well as the modeling of the permanent contact contribution to the particulate pressure. The implementation of the dense granular flow rheology, the mixing length and the two-phase $k-\omega$



**Table 11.** Summary of the numerical results obtained for the scour at an apron using the different combinations of turbulence and granular stress models and comparison with existing two-phase numerical results on this configuration.

| Case | $max(\delta_s/h_0)$ | $n_s$ | $T_s$ (s) | $\zeta_{s0}^{\infty}$ (degrees) | $T_{\zeta_{s0}}$ (s) |
|---|---|---|---|---|---|
| Amoudry et al. (2008) | 0.2 | 0.56 | 600 | -11.4 | 4 |
| Cheng et al. (2017) | 0.16 | 0.54 | 1100 | -14.55 | 15.2 |
| k-$\varepsilon$ + KT | 0.111 | 0.530 | 3896 | -19.18 | 9.42 |
| k-$\varepsilon$ + $\mu(I)$ | 0.089 | 0.472 | 11006 | -11.23 | 3.89 |
| k-$\omega$ + KT | 0.105 | 0.51 | 5572 | -18.16 | 9.45 |
| k-$\omega$ + $\mu(I)$ | 0.089 | 0.436 | 16224 | -10.79 | 2.9 |

models are original contributions. The dense granular flow rheology is implemented using a regularization technique and is verified against an analytical solution for the laminar bed-load problem. The application of the model to turbulent sheet flows allows us to discuss the sensitivity of the model results to different combinations of inter-granular stress and turbulence models. A first set of tuning coefficient values is provided, and the results are in reasonable agreement with four different experimental configurations. The last application on scour allows to illustrate the multi-dimensional capabilities of the solver. The scaling laws proposed by earlier works is recovered by the model but the results are also sensitive to the modeling choices on the granular and turbulence models. In light of these model applications, some questions remain on the optimum values of the turbulence model coefficients, which will need more high resolution measurements, for a wide range of flow conditions. The open-source numerical model presented here is expected to facilitate this endeavor in the future.

As a general conclusion, the aim of this contribution is to provide a comprehensive two-phase flow sediment transport modeling framework to the scientific community. Intense efforts have been made to ensure its reliability and numerical robustness. This numerical tool is suitable to address various physical problems for which the classical sediment transport approach is not working very well or require more model assumptions. However, the readers are reminded that that two-phase flow simulations are still relatively time consuming and require finer spatial resolution and smaller time steps than classical sediment transport models. We encourage more contributions to the model development from the community effort, and we will be delighted to integrate them in the future releases of sedFoam.

## 7 Code availability

The code is available at http://github.com/SedFoam/sedfoam and the python package for postprocessing of the tutorials is available at http://bitbucket.org/sedfoam/fluidfoam.





**Acknowledgment**

Julien Chauchat, Tim Nagel and Cyrille Bonamy are supported by the Region Rhones-Alpes (COOPERA project and Explora Pro grant), the French national programme EC2CO-LEFE MODSED. Zhen Cheng and T.-J Hsu are supported by National Science Foundation (OCE-1537231; OCE-1635151) and Office of Naval Research (N00014-16-1-2853) of USA.

5      Numerical simulations were carried out on MILLS/FARBER at the University of Delaware and on the Froggy platform of the CIMENT infrastructure (https://ciment.ujf-grenoble.fr), which is supported by the Rhone-Alpes region (GRANT CPER07-13 CIRA) and the Equip@Meso project (reference ANR-10-EQPX-29-01) of the programme Investissements d'Avenir supervised by the Agence Nationale pour la Recherche.

The authors would also like to acknowledge the support from the program on "Fluid-Mediated Particle Transport in Geo-

10    physical Flows" at the Kavli Institute for Theoretical Physics, Santa Barbara, USA. The laboratory LEGI is part of the LabEx Tec 21 (Investissements d'Avenir - grant agreement nANR-11-LABX-0030) and Labex OSUG@2020 (ANR10 LABX56).

We are grateful to the developers involved in OpenFOAM which are the foundation of the model presented in this paper.





# 1  Notations

| | |
|---|---|
| $C_d$ | drag coefficient (-) |
| $d_{eff}$ | effective particle diameter (m) |
| $D_{small}$ | regularisation parameter for the rheology ($\text{s}^{-1}$) |
| $D_\Theta$ | conductivity of granular temperature (kg $\text{m}^{-1}\,\text{s}^{-1}$) |
| $f_i$ | external force that drives the flow (kg $\text{m}^{-2}\,\text{s}^{-2}$) |
| $g_{s0}$ | radial distribution function for dense rigid spherical particles gases (-) |
| $h_b$ | seabed height (m) |
| $h_0$ | water column height (m) |
| $I$ | inertial number (-) |
| $I_v$ | viscous number (-) |
| $I_0$ | empirical parameter of the granular rheology (-) |
| $J_{int}$ | energy dissipation (or production) due to the interaction with the carrier fluid phase(-) |
| $k$ | turbulent kinetic energy ( $\text{m}^2\,\text{s}^{-2}$) |
| $k_s$ | Nikuradse roughness length (m) |
| $K$ | drag parameter (kg $\text{m}^{-3}\,\text{s}^{-1}$) |
| $n_s$ | characteristic exponent (-) |
| $p$ | fluid pressure (Pa) |
| $\tilde{p}^a$ | particle normal stress (Pa) |
| $p^a$ | collisional component of the particle pressure (Pa) |
| $p^{ff}$ | permanent contact component of the particle pressure (Pa) |
| $q_j$ | flux of granular temperature (kg $\text{s}^{-1}$) |
| $R_{ij}^{bt}$ | Reynolds stress (Pa) |
| $Re_p$ | Particulate Reynolds number (-) |
| $r_{ij}^{b}$ | viscous stress (Pa) |
| $S_f$ | energy slope (%) |
| $S_{ij}^k$ | deviatoric part of the phase k strain rate tensor (Pa) |
| $S_t$ | Stokes number (-) |
| $t$ | time (s) |
| $T_s$ | scour characteristic timescale (s) |
| $T_{\alpha_{s0}}$ | equilibrium timescale for upstream bed angle (s) |
| $t_{mf}$ | turbulent drag parameter (-) |



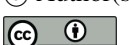

| $t_l$ | characteristic time scale of energetic eddies (s) |
| $t_p$ | particle response time (s) |
| $u_i^k$ | phase k velocity, $i = 1; 2; 3$ represents streamwise, spanwise and vertical component, respectively ( $\mathrm{m\,s^{-1}}$ ) |
| $u_*$ | bed friction velocity ( $\mathrm{m\,s^{-1}}$ ) |
| $\bar{U}$ | mean velocity ( $\mathrm{m\,s^{-1}}$ ) |
| $W_{fall}$ | settling velocity ( $\mathrm{m\,s^{-1}}$ ) |
| $Y_i^{low}$ | position of the upper sediment interface for pure sedimentation test case ( m ) |
| $Y_i^{up}$ | position of the upper sediment interface for pure sedimentation test case ( m ) |

Table 12: Nomenclature.

*Greek symbols*

| $\alpha$ | solid phase sediment concentration (-) |
| --- | --- |
| $\alpha^0$ | initial solid sediment concentration (-) |
| $\alpha_{s0}^\infty$ | upstream bed angle at equilibrium in the scour at an apron case (degrees) |
| $\beta$ | fluid phase volume concentration (-) |
| $\delta_s$ | scour depth (m) |
| $\Delta t$ | time step (s) |
| $\varepsilon$ | turbulent kinetic energy dissipation rate ($\mathrm{m^2\,s^{-3}}$) |
| $\gamma$ | energy dissipation rate due to inelastic collision ($\mathrm{kg\ m^{-1}\,s^{-3}}$) |
| $\kappa$ | Von Karmann constant (-) |
| $\lambda$ | bulk viscosity ($\mathrm{kg\ m^{-1}\,s^{-1}}$) |
| $\mu$ | friction coefficient |
| $\mu_c^a$ | particle shear viscosity ($\mathrm{kg\ m^{-1}\,s^{-1}}$) |
| $\mu_s$ | static friction coefficient for the rheology (-) |
| $\mu_2$ | dynamic friction coefficient for the rheology (-) |
| $\nu_{Fr}^a$ | frictional viscosity ( $\mathrm{m^2\,s^{-1}}$ ) |
| $\nu_{Eff}^k$ | phase k effective viscosity ( $\mathrm{m^2\,s^{-1}}$ ) |
| $\nu_t^b$ | turbulent viscosity ( $\mathrm{m^2\,s^{-1}}$ ) |
| $\nu^b$ | fluid viscosity ( $\mathrm{m^2\,s^{-1}}$ ) |
| $\nu^{mix}$ | mixture viscosity ( $\mathrm{m^2\,s^{-1}}$ ) |
| $\omega$ | turbulent kinetic energy specific dissipation rate ($\mathrm{s^{-1}}$) |
| $\sigma_k$ | turbulent coefficients (-) |
| $\sigma_\omega$ | turbulent coefficients (-) |





| | | |
|---|---|---|
| $\Theta$ | granular temperature ( $\mathrm{m^2\,s^{-2}}$) | |
| $\tau_{ij}^{b}$ | fluid stress (Pa) | |
| $\tau_{ij}^{a}$ | particle shear stress (Pa) | |
| $\tau_{ij}^{ac}$ | collisional stress (Pa) | |
| $\tau_{ij}^{af}$ | frictional stress (Pa) | |
| $\zeta_{s0}$ | upstream bed angle in the scour at an apron case (degrees) | |

| symbols | keywords | default value | description |
|---|---|---|---|
| $\alpha_{min}^{Fric}$ | alphaMinFriction | – | minimum friction packing volume concentration (-) |
| $\alpha_{max}$ | alphaMax | – | maximum packing volume concentration (-) |
| $\alpha_{small}$ | alphaSmall | $1 \times 10^{-4}$ | minimum friction packing volume concentration (-) |
| B | B | 0.25 | empirical coefficient (-) |
| $B_\phi$ | Bphi | 1/3 | parameter of the dilatancy law (-) |
| $C_\mu$ | Cmu | 0.09 | turbulent coefficient (-) |
| $C_{3\varepsilon}$ | C3ep | 1.2 | $k - \varepsilon$ turbulent coefficients (-) |
| $C_{4\varepsilon}$ | C4ep | 0 | $k - \varepsilon$ turbulent coefficients (-) |
| $C_{3\omega}$ | C3om | 0.2 | $k - \omega$ turbulent coefficients (-) |
| $C_{4\omega}$ | C4om | 0 | $k - \omega$ turbulent coefficients (-) |
| $d$ | d | – | sediment particle diameter (m) |
| $e$ | e | – | coefficient of restitution during the collision (-) |
| $\eta_0$ | eta | – | particle phase stress empirical coefficient (-) |
| $\eta_1$ | p | – | particle phase stress empirical coefficient (-) |
| $Fr$ | Fr | – | particle phase stress empirical coefficient (-) |
| $g_i$ | g | – | gravitationnal acceleration ( $\mathrm{m\,s^{-2}}$) |
| $h_{Exp}$ | hExp | 1.0 | hindrance exponent (-) |
| $\kappa$ | kappaLM | 0.4 | on Karmann constant (-) |
| $\rho^a$ | rho | – | density of phase k ($\mathrm{kg\,m^{-3}}$) |
| $\rho^b$ | rho | – | density of phase k ($\mathrm{kg\,m^{-3}}$) |
| $S_{US}$ | SUS | 1.0 | inverse Schmidt number (-) |
| $\theta_f$ | phi | – | sediment angle of repose (degrees) |
| $\psi$ | sF | 1.0 | grain shape factor (-) |

Table 14: optional input coefficients



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
