# Peer review of "SedFoam-2.0: a 3D two-phase flow numerical model for sediment transport"

_Geoscientific Model Development, 2017_

## Referee Comment (RC1) · Anonymous Referee #1 · 6 Jul 2017

General comments

This paper presents a 3D two-phase flow numerical model for sediment transport (SedFoam-2.0) in detail, including the mathematical formulation and the numerical implementations. The authors newly include the mixing length turbulence model, the $k - \omega$ model, and dense granular flow rheology into SedFoam. The main purpose is to provide a comprehensive numerical framework that solves the two-phase flow equations in three dimensions with the capability to select different combinations of turbulent model and granular stress model for sediment transport. This paper is well written and pleasant to read. The reviewer suggests acceptance after minor revisions.

Specific comments

[Figure]

Page 5: The authors state that $h_{Exp}$ depends on the particulate Reynolds number. Its default value is 2.65 in SedFoam-2.0. To avoid misuse by users, it should be mentioned the range of the particulate Reynolds number in which $h_{Exp} = 2.65$ is applicable.

Page 14: Is the value of $B_{phi}$ in Eq. (46) 1/3 as that in Eq. (43)?

Page 23, Figure 2: Lee et al. (2016) mentioned that SedFoam might yield fluctuating particle pressure. From Fig. 2, SedFoam-2.0 seems improve this weakness. How does sedFoam-2.0 improve this weakness?

Technical corrections

Page 11, Line 8: Double "are."

Page 24, Line 12: Change "$210^{-4}$" to "$2 \times 10^{-4}$."

Reference

Lee C-H, Low YM, Chiew Y-M. Multi-dimensional rheology-based two-phase model for sediment transport and applications to sheet flow and pipeline scour. Phys Fluids 2016;28:53305. doi:10.1063/1.4948987.

---

## Referee Comment (RC2) · Anonymous Referee #2 · 10 Jul 2017

The authors present a three-dimensional two-phase flow numerical model for sediment transport. This type of model requires closure schemes, not only for fluid turbulence but also for particle phase stresses. The model introduced in this paper includes a series of closure schemes for each, all within a single numerical framework. This particular aspect is a welcome addition to the two-phase sediment transport modelling community. A number of models have been introduced over the past 20 years, using slightly different assumptions, closures, simplifications, and intercomparison has remained elusive and difficult, partly due to the lack of a common numerical platform. The result is that there does not seem to be a widespread consensus on the "best practice" for these models.

The authors present the mathematical framework of the model in sufficient details.

[Figure]

However, I believe it would be useful to include a brief discussion summarising how the present formulations compare to existing published models. While this can be somewhat inferred from the text and table 1, it would probably benefit from a few summary sentences that include a brief discussion of potential implications of the differences. A particular aspect to consider would be whether all models using kinetic theory per table 1 are using the exact same model and how these relate to the kinetic theory introduced in the paper?

The authors also present the numerical implementation well, but I would have expected some mention of constraints on the time step. For example, is there a dynamic adjustment of the time step, as in some earlier models, or a static time step, in which case it would be good to discuss how to set this in the first place?

The overall model is then tested in four specific cases: either benchmarks or applications. The ability of the model to use different closures is well used in these benchmarks/applications to gain insight on best practice for two-phase sediment transport modelling. I, however, did not fully understand the reason behind the split behind benchmark and application. If the reasoning is that the benchmark tests serve as validation of the model, then only a few components are truly validated.

This paper does have some issues that would need to be addressed before publication. The most important aspect is that rationale(s) and justification(s) for the work undertaken are rather weak throughout the manuscript and should be improved.

1) I find the argumentation presented in the introduction to be misguided and I think the introduction needs significant revision. Yes, sediment transport is important and "a major societal issue for the management of natural systems". Yes, we require "the development of comprehensive models". However, the authors fail to fully recognise what modelling approaches are actually being used to inform coastal management now and in the foreseeable future, i.e. probably not two-phase sediment transport modelling. The reason is that it still is unrealistic to scale up two-phase sediment

transport model to the spatio-temporal scales of interest to coastal managers under the typically preferred approaches that emphasise probabilistic hazard assessment. It is important to stress here that I am not stating that two-phase sediment transport modelling is not useful and important, just not for the overall rationale postulated in the introduction. If the authors want to relate their effort to "better coastal modelling", I believe that an argument revolving around improving representation(s) of detailed complex physical processes in the models used for coastal management would be far more convincing.

2) The authors use several times "motivated by [previous publication]" as justification to include some specific closures or focus on a specific case study. I find these rather weak justifications and would encourage the authors to think about how physical processes are reproduced instead. For example, a reason to introduce a k-omega model could be because of known deficiencies of the k-epsilon model (e.g. pressure gradients).

3) The rationale for the choice of the four benchmark tests and applications is not evident. I note that only one case out of four actually uses "real" sediment. Why and would there be implications for applicability of the model? I also note that only steady sheet flow is looked into, even though wave-generated sheet flows have been a key application of two-phase flow modelling, and would be a very important application given the shortcomings of the single phase approach as stated in the introduction. Again, why this choice? Finally, while a two-dimensional case is indeed presented and the formulation of the model is implicitly in 3D (as well as the code as presume), I note that no three-dimensional validation or application is presented. On these points, I fully recognise that adding more cases may not be feasible within the scope of the paper, in which case some discussion of the points raised above would be needed.

Specific comments:

Page 5, line 9: What about other forces than drag? I think some of the early theoretical

works on multiphase governing equations mention other forces such as added mass and lift.

Page 6, line 1: should read "fluid stresses consist of . . .."

Page 8, equation 19: I believe it would be good to mention the work by W Kranenburg testing different equations. (Kranenburg et al., 2014, Advances in Water Resources)

Page 22, line 4: why using a Coulomb rheology?

Page 22, line 20: this is because the sediment . . .

Page 23: figure 2: Please discuss the discrepancy in the middle bottom panel

Page 24, line 6: can sheet flows not be turbulent?

Page 26, line 18-21: I think this paragraph would be better as part of the rationale for the work.

Figures 3 and 4: The figures are not clear enough. There are more line types in the plots than in the legend, the different colour are not explained.

Page29, line 3: studies of . . ..

---

## Short Comment (SC1) · 21 Jul 2017

In order to maintain a high degree of reproducibility the model version referenced in a manuscript need to be marked as a release in GitHub if this is the mechanism to make the code accessible. Commits into the repository after submission of the paper can compromise reproducibility if it is not clearly documented which status of the repository is discussed in the manuscript. GMD is strongly encouraging (but does not enforce) authors to provide persistent access to their program code and data used in the manuscript. Typically this is guaranteed through the use of a DOI which can be created for releases made in GitHub using Zenodo. This DOI should be cited in the paper.

[Figure]

The author should also clearly state in the paper which version of OpenFOAM has been used and add a brief statement on the licence under which sedfoam can be used.

Many thanks

Lutz Gross GMD Editor

---

## Author Comment (AC1) · 31 Jul 2017

Dear Editor,

Thank you for the comment, we were waiting for the reviewer's feedback before submitting the source code on Zenodo. As we believe the source code won't be changed before final acceptance we did as you requested, the Github repository has been tagged as V2.0

https://github.com/SedFoam/sedfoam/releases/tag/v2.0

and this version of the code is now registered in Zenodo as

https://zenodo.org/record/836643
with the following DOI

http://dx.doi.org/10.5281/zenodo.836643

The manuscript has been updated with these informations and the OpenFoam version (2.4.0) used to compile the source code is also specified in the section Code availability.

Best regards,

Julien Chauchat

---

## Author Comment (AC2) · 31 Jul 2017

The full answer is also provided as a pdf with different colors for reviewer comments and authors answers.

Anonymous Referee #1

General comments This paper presents a 3D two-phase flow numerical model for sediment transport (SedFoam-2.0) in detail, including the mathematical formulation and the numerical implementations. The authors newly include the mixing length turbulence model, the k$\omega$ model, and dense granular flow rheology into SedFoam. The main purpose is to provide a comprehensive numerical framework that solves the two-phase flow equations in three dimensions with the capability to select different combinations

of turbulent model and granular stress model for sediment transport. This paper is well written and pleasant to read. The reviewer suggests acceptance after minor revisions.

Specific comments

Page 5: The authors state that hExp depends on the particulate Reynolds number. Its default value is 2.65 in SedFoam-2.0. To avoid misuse by users, it should be mentioned the range of the particulate Reynolds number in which hExp= 2.65 is applicable.

This parametrization is not supposed to be very sensitive, according to Di Felice (1994) the value of hExp depends and the particulate Reynolds number. For particulate Reynolds number between 1 and 300, which is mostly the case for sediment transport applications, the value of hExp varies between 2 and 2.65. The following paragraph has been added to the manuscript between l. 11 and l. 16 p. 6:

"The hindrance function $\beta^{-h_{Exp}}$ represents the drag increase when the particle volume concentration increases. $h_{Exp}$ is the hindrance exponent that depends on the particulate Reynolds number \cite{difelice1994}. For simplicity, the value of $h_{Exp}$ is assumed to be constant (default value is 2.65), and its value can be specified from the \textit{constant/transportProperties} file in \sedFoam. This hypothesis is valid for particulate Reynolds numbers lower than unity or larger than 300, within this range the exponent value decreases down to $h_{Exp}\approx2$."

Page 14: Is the value of Bphi in Eq. (46) 1/3 as that in Eq. (43)?

There was a mistake in the manuscript about this value which is now corrected, the value of Bphi for the viscous regime of the granular rheology is 1 Eq. (43) and between 0.31 and 2/3 for the grain inertia regime Eq. (46). This has now been corrected in the manuscript.

Page 23, Figure 2: Lee et al. (2016) mentioned that SedFoam might yield fluctuating particle pressure. From Fig. 2, SedFoam-2.0 seems improve this weakness. How does sedFoam-2.0 improve this weakness?

[Figure]

Following the idea suggested by Lee et al. (2016) we have used the fixedFluxPressure boundary condition for the pressure at the bed. We have added the citation to Lee et al. (2016) and we apologize for not having cited this paper in the first version.

Technical corrections Page 11, Line 8: Double "are." Page 24, Line 12: Change "210ˆ{-4}" to "2×10ˆ{-4}"

Typos have been corrected.

Reference Lee C-H, Low YM, Chiew Y-M. Multi-dimensional rheology-based two-phase model for sediment transport and applications to sheet flow and pipeline scour. Phys Fluids 2016;28:53305. doi:10.1063/1.4948987.

Please also note the supplement to this comment:
https://www.geosci-model-dev-discuss.net/gmd-2017-101/gmd-2017-101-AC2-supplement.pdf

---

## Author Comment (AC3) · 1 Aug 2017

The full answer is also provided as a pdf with different colors for reviewer comments and authors answers.

Anonymous Referee #2

 The authors present a three-dimensional two-phase flow numerical model for sediment transport. This type of model requires closure schemes, not only for fluid turbulence but also for particle phase stresses. The model introduced in this paper includes a series of closure schemes for each, all within a single numerical framework. This particular aspect is a welcome addition to the two-phase sediment transport modelling community. A number of models have been introduced

over the past 20 years, using slightly different assumptions, closures, simplifications, and intercomparison has remained elusive and difficult, partly due to the lack of a common numerical platform. The result is that there does not seem to be a widespread consensus on the "best practice" for these models. The authors present the mathematical framework of the model in sufficient details.

However, I believe it would be useful to include a brief discussion summarising how the present formulations compare to existing published models. While this can be somewhat inferred from the text and table 1, it would probably benefit from a few summary sentences that include a brief discussion of potential implications of the differences. A particular aspect to consider would be whether all models using kinetic theory per table 1 are using the exact same model and how these relate to the kinetic theory introduced in the paper?

The sediment transport models listed in Table 1 adopted earlier version of kinetic theory for dry granular flow (e.g., Jenkins and Savage 1983) that does not consider streaming effect due to particle-particle interaction (e.g. Lun and Savage 1987) and the effect of fluid phase. For sediment transport application, the streaming effect provides a smooth decay of granular temperature in the dilute region and ensures a more stable transition of suspension mechanisms from particle collision to turbulent suspension. Hence, in the present model, we adopt the later version of the kinetic theory suggested by Ding and Gidaspow (1990), which include streaming effects and the effect of fluid phase.

The following paragraph has been added to the introduction (from l.22 p. 3 to l. 9 p. 4):

"In this approach, the particle stress associated with particle-particle collisions are modeled by the fluctuation energy of the particle phase (or granular temperature). Various models were developed to model the granular temperature. \cite{Jenkins1998} first applied kinetic theory for dry granular flows to sheet flow, and the granular temperature transport equation was later extended to consider the fluid-sediment turbulence interactions \cite[][e.g.]{hsu2004b,chauchat2008}. The aforementioned kinetic theory

considered dense collisions of particles, while the streaming effect of particle random motions were missing in the dilute concentration regime. A further extension to include the streaming effects were developed by \cite[][e.g.]{Lun1987,Ding1990}, and it was implemented into a more complete two-phase model by \cite{cheng2017}. In contrast of solving the transport of granular temperature, \cite{jha2010} used a mixing length concept for the particle phase, and a simpler algebraic model for the granular temperature was used with success."

The authors also present the numerical implementation well, but I would have expected some mention of constraints on the time step. For example, is there a dynamic adjustment of the time step, as in some earlier models, or a static time step, in which case it would be good to discuss how to set this in the first place?

Yes the time step can be adjusted dynamically using different Courant numbers based on the averaged fluid phase velocities, the relative velocity and the interfacial velocity. This is now written in the manuscript:

"The time step, $\Delta t$, can be adjusted automatically based on two Courant numbers, one related to the local flow velocity and the local grid size (the same as for single phase problems) and one related to the relative velocity which is specific to the coupling of the fluid and sediment phase momentum equations in the two-phase flow model. Our practice is to set these two Courant numbers to 0.3 and 0.1, respectively."

The overall model is then tested in four specific cases: either benchmarks or applications. The ability of the model to use different closures is well used in these benchmarks/ applications to gain insight on best practice for two-phase sediment transport modelling. I, however, did not fully understand the reason behind the split behind benchmark and application. If the reasoning is that the benchmark tests serve as validation of the model, then only a few components are truly validated.

Following the reviewer comment we decided to combine former sections 4 and 5 into a single section 4 called "Model verification and benchmarking".

This paper does have some issues that would need to be addressed before publication. The most important aspect is that rationale(s) and justification(s) for the work undertaken are rather weak throughout the manuscript and should be improved.

1) I find the argumentation presented in the introduction to be misguided and I think the introduction needs significant revision. Yes, sediment transport is important and "a major societal issue for the management of natural systems". Yes, we require "the development of comprehensive models". However, the authors fail to fully recognise what modelling approaches are actually being used to inform coastal management now and in the foreseeable future, i.e. probably not two-phase sediment transport modelling. The reason is that it still is unrealistic to scale up two-phase sediment transport model to the spatio-temporal scales of interest to coastal managers under the typically preferred approaches that emphasise probabilistic hazard assessment. It is important to stress here that I am not stating that two-phase sediment transport modelling is not useful and important, just not for the overall rationale postulated in the introduction. If the authors want to relate their effort to "better coastal modelling", I believe that an argument revolving around improving representation(s) of detailed complex physical processes in the models used for coastal management would be far more convincing. We agree with the reviewer's point and in the revised manuscript, we had revised the introduction to better reflect the state-of-the-art in morphodynamic modeling and clarify the goal of this study (see line 29 p. 2 to line 2 p. 3)

"Addressing these issues requires the development of comprehensive models that account for the variety of complex hydrodynamics and sediment transport processes in a regional-scale setting (e.g., Lesser et al. 2004; Roelvink et al. 2009). Because sediment transport occurs very close to the bed, effective parameterizations of sediment transport are needed. In the past decade, significant progresses have been demonstrated to utilize detailed numerical model to understand sheet flow processes and effective parameterizations for coastal sediment transport have been developed (van der A et al. 2013). To further tackle more complex sediment transport problems,

such as bedform evolution, scour, bank erosion and dune erosion, further expansion of these models through a community effort is urgently needed. The development of more comprehensive sediment transport models integrating the complexity of the underlying physical coupling mechanisms is the main goal of the open-source community model presented herein."

2) The authors use several times "motivated by [previous publication]" as justification to include some specific closures or focus on a specific case study. I find these rather weak justifications and would encourage the authors to think about how physical processes are reproduced instead. For example, a reason to introduce a k-omega model could be because of known deficiencies of the k-epsilon model (e.g. pressure gradients).

We would like to thank the reviewer for this comment, indeed, the justification for the implementation of the k-omega model and for the choice of the different test cases was not precise enough and we have modified the introductory par the following subsections:

k-omega "It is well known that the original $k-\epsilon$ model has been derived for high Reynolds number flows and is not very accurate to describe transitional flows such as the situation of the flow reversal in a wave boundary layer\citep{guizien2001}. For this situation and for near wall treatment the $k-\omega$ model is more suitable and more stable than the $k-\epsilon$ model \citep{guizien2001}. Another physical situation in which a $k-\omega$ model works better than a $k-\epsilon$ model is in the presence of an adverse pressure gradient such as the downward facing step or at the upstream side of an obstacle \citep{menter1993,wilcox2006}. In order to test the influence of the turbulence model, a two-phase $k-\omega$ model is introduced in the present contribution which is very similar to \cite{jha2009,amoudry2014} ones."

Test case 1 "The first test case corresponds to a pure sedimentation of non-cohesive particles, this test case allows to validate the implementation of the pressure-velocity

coupling algorithm when the flow is induced by the sediment phase. The other component that is tested here is the permanent contact pressure model (eq. \ref{eq:paff}) for $p^{ff}$ that allows to predict a stable deposited sediment bed."

Test case 2 "The second test case is inspired by \cite{chauchat2010} in which an analytical solution for laminar bed-load driven by a Poiseuille flow has been used to verify a three-dimensional numerical model. The goal of this test case is to verify the numerical implementation of the granular rheology in \sedFoam. The novelty compared with \cite{chauchat2010} is that the solid phase concentration is obtained as a result of the model in \sedFoam however it was imposed constant in our earlier work."

Test case 3 "In this subsection the model results are compared with experimental results from \cite{revil-baudard2015} and \cite{sumer1996} for turbulent sheet flows, the goal of these test cases is to validate the numerical implementation of different turbulence model (mixing length and $k-\omega$) and to calibrate the free parameter $B$."

3) The rationale for the choice of the four benchmark tests and applications is not evident. I note that only one case out of four actually uses "real" sediment. Why and would there be implications for applicability of the model? I also note that only steady sheet flow is looked into, even though wave-generated sheet flows have been a key application of two-phase flow modelling, and would be a very important application given the shortcomings of the single phase approach as stated in the introduction. Again, why this choice? Finally, while a two-dimensional case is indeed presented and the formulation of the model is implicitly in 3D (as well as the code as presume), I note that no three-dimensional validation or application is presented. On these points, I fully recognise that adding more cases may not be feasible within the scope of the paper, in which case some discussion of the points raised above would be needed.

As the model is based on physical ground it should work for both lightweight sediments and sand. The point is that there is no experimental data for unidirectional sheet

flow providing velocity and concentration profiles for the same experimental condition. Concerning oscillatory sheet flows using sand, it has been addressed by Cheng et al (2017) using the same numerical model (Kinetic theory and k-epsilon only) and we do not think it is needed to repeat these results here. This is now explained in the introduction of the paper. The following sentence has been added to the introduction (l. 12-13 p. 4)

" \cite{cheng2016} have applied sedFoam using the kinetic theory of granular flows and the $k$-$\epsilon$ turbulence model to reproduce oscillatory sheet flows of fine, medium and coarse sand \cite{odonoghue2004}."

Concerning the 3D nature of the numerical model, it is indeed 3D but, as the reviewer mentioned, the two-phase flow model is quite computationally demanding and we preferred to focus on 2D configurations that already give an overview of the multi-dimensional capabilities of the code. 3D configurations are under development and will be the subject of future papers.

We would like to remind here that the primary goal of this paper is to present, in details, the mathematical and numerical model formulation of model to serve as a documentation for future users. We agree with most of what the reviewer ask but we believe that it goes beyond the scope of the present paper. We did our best to address the major points and we hope that the reviewer will find our answers satisfactory.

Specific comments: Page 5, line 9: What about other forces than drag? I think some of the early theoretical works on multiphase governing equations mention other forces such as added mass and lift.

Other forces than drag such as added mass or lift forces might also play a role but in most two-phase flow models only drag force is accounted for. According to Jha and Bombardelli (2010) the lift force only accounts for less than 4% of the drag force and it has no significant effect on the results. The added mass force could play a non negligible role in the denser part of the flow and this would deserve further investigation.

The following sentence has been added to the manuscript (l. 1-6 p. 6):

"Other forces such as the lift force or the added mass force could play a role in sediment transport, according to \cite{jha2010}, the lift force in dilute suspended sediment transport only represent 4\% of the drag force and the added mass force can be of order of 10\% in the near bed region. The influence of the added mass force would require further investigation that are beyond the scope of the present paper. "

Page 6, line 1: should read "fluid stresses consist of ..." Done

Page 8, equation 19: I believe it would be good to mention the work by W Kranenburg testing different equations. (Kranenburg et al., 2014, Advances in Water Resources) Done

Page 22, line 4: why using a Coulomb rheology? The Coulomb rheology is used because it is the only way to have an analytical solution of the problem that allows to actually validate the numerical implementation of the granular rheology.

Page 22, line 20: this is because the sediment : Done

Page 23: figure 2: Please discuss the discrepancy in the middle bottom panel There is no discrepancy here, the results obtained using sedFoam with the mu(I) rheology as to be compared with the numerical solution using the mu(I) rheology (in green) and not with the black dotted line that represent the analytical solution using the Coulomb rheology.

Page 24, line 6: can sheet flows not be turbulent? We have added turbulent before sheet flows to differentiate with the previous laminar bedload case that could also be called sheet flows.

Page 26, line 18-21: I think this paragraph would be better as part of the rationale for the work. Done

Figures 3 and 4: The figures are not clear enough. There are more line types in the

plots than in the legend, the different colour are not explained. The two line colors represent fluid velocity in blue and sediment velocity in red. In order to limit the number of profiles plotted we have replaced the two profiles per case by the volume averaged velocity profile computed asĂă: U = alpha Ua + beta Ub. This is now explained in the manuscript as well.

Page29, line 3: studies of ... Done

Please also note the supplement to this comment:
https://www.geosci-model-dev-discuss.net/gmd-2017-101/gmd-2017-101-AC3-supplement.pdf

––––––––––––––––––––––––––––––

---

## Author Response (AR2)

**Topical Editor Decision: Publish subject to minor revisions (Editor review)** (21 Sep 2017) by James R. Maddison Comments to the Author: I believe most reviewer comments have been addressed. However please review some comments from reviewer 2:

- Page 22, line 4: why using a Coulomb rheology?

The justification could appear in the text.

The introduction of the test case has been rephrased to justify the use of the Coulomb rheology :

« The goal of this test case is to verify the numerical implementation of the granular rheology in sedFoam against an analytical solution. The analytical solution can derived only when using the Coulomb rheology (see table  $ref{MuIModel}$ ) and the Einstein mixture fluid viscosity model (see table  $ref{FluidViscosityModel}$ ). Those parameterization will be used first for verification, then the numerical solution using the mu(I) rheology will compared with previous numerical solution. »

- Page 23: figure 2: Please discuss the discrepancy in the middle bottom panel

There is a discrepancy between the numerical and analytical solutions -- can this be explained?

There is no discrepancy, the rheology is different between the analytical solution and the sedFoam result. The analytical solution is just presented for reference. As it might not be very clear, the analytical solution has been removed from figure 2b.

- Figures 3 and 4: The figures are not clear enough. There are more line types in the plots than in the legend, the different colour are not explained.

The figure presentation should be improved. The figures make use of colour without explanation. The black and blue elements of the figures are unlabelled. It should be clarified which are the experimental values for comparison.

The captions have been updated to explain the colors. No labels has been added to the figure to avoid having too much labels.

I also have the following comments:

x- Abstract: The final two sentences now appear inconsistent with the revised text.

The abstract has been updated, the new sentences now reads :

« The numerical implementation is demonstrated on four test cases, sedimentation of suspended particles, laminar bed-load, sheet-flow and scour at an apron. These test cases illustrate the capabilities of \sedFoam~ to deal with complex turbulent sediment transport problems with different combinations of inter-granular stress and turbulence models. »

- Page 3 lines 2-5: The references to "community" efforts seem unnecessary here. This is referenced later at the bottom of page 4.

The sentence has been deleted.

- Page 4 lines 29-32: Please avoid the use of the first person here. The final sentence is subjective and could be removed.

The last sentence has been removed.

- Page 6 lines 20-22: Please add a justification or reference for the validity of this hypothesis.

The choice of the drag model is justified and the following sentences have been modified :

« This drag model can be chosen by the keyword ``GidaspowSchillerNaumann" in the file \textit{constant/interfacialProperties} and it is especially well adapted for dealing with suspended particles. For situation in which the fluid flow inside the porous sediment bed has to be accurately solved, other drag models are available in \sedFoam but as they are not relevant to the test cases investigated in this paper, their description is omitted. »

- Page 8 line 10: Please add further details regarding the clipping.

The clipping is now clearly stated in the manuscript :

« A special treatment of the term  $(1-\lambda)/(n)^{-n}\$  is needed to avoid dividing by zero as  $\lambda = 1-\min(\lambda)^{-n}$ , basically, this term is clipped as follows  $1-\lambda = 1-\min(\lambda)^{-n}$ .

- Tables 2, 3, 6, 7, 8: I find these very unclear. Please clarify how these should be interpreted -- it is unclear what is being set (parameter values?), particularly in table 3.

Concerning table 3, the parameters mentioned corresponds to the input keywords that can be used for choosing the turbulence model. The captions of the tables have been rephrased to make this point more clear and the parameters that has to be set for each model is now specified in the table as well. This table has been moved at the end of the turbulence model subsection.

For tables 2, 6, 7 and 8, the table captions and their content has been improved.

- There are references to a non-existent "appendix 13".

This has been corrected in the manuscript.

- Page 20 lines 20-23: Please clarify how the two Courant numbers are used. Are these used to bound the maximum timestep?

- Page 21 line 18, page 24 line 13: The use of a fixed timestep seems to conflict with the setting of a Courant number on page 20. Please clarify.

The Courant numbers are now defined in the manuscript and their use for setting the time step in adjustable mode is explained :

« The time step,  $\Delta t$ , can be set to a constant value or adjusted automatically based on two Courant numbers, one related to the local flow velocity and the local grid size  $Co=1/2 \sum_f \Phi_f \Delta t / V_p$  (the same as for single phase problems) and one related to the relative velocity  $Co_r=1/2 \sum_f \B_{r}^{-1/2} \sum_f \B_{r}^{-1/2} \buildrel f$

coupling of the fluid and sediment phase momentum equations in the two-phase flow model. The most limiting time step is used as the criterion for setting the adjustable time step. Our practice is to set these two Courant numbers to 0.3 and 0.1, respectively. »

As is know made more clear in the manuscript, adjustable time-step can be used or not in which case the time step is fixed at a constant value.

- There are a number of places which seem to refer to SedFoam-2.0 parameter values (e.g. page 24 lines 12, 16, page 26 line 15, page 31 line 15, table 10 caption). These are potentially confusing for readers unfamiliar with the details of the model configuration. Please clarify (e.g. "the pressure gradient is fixed by setting the model parameter gradPMEAN=100 kg m^{-2} s^{-2}").

The meaning of the pressure gradient gradPMEAN is now clarified and an external volume force denoted as \$f\$ is introduced in the momentum equations. For the parameters appearing at page 12,16, these are the fluid and particle densities and fluid viscosity for which we don't feel clarification is required but maybe we misunderstood the editor's comment?

For the caption of table 10, the correspondance between the openFOAM boundary condition names and classical boundary conditions (Dirichlet and Neumann) is now given :

« The following abbreviations are used:  $zG = \text{textit}\{\text{zeroGradient}\}\$  for Neumann boundary condition,  $fV = \text{textit}\{\text{fixedValue}\}\$  for Dirichlet boundary conditions,  $dM = \text{textit}\{\text{directionMixed}\}\$  for mixed Dirichlet-Neumann boundary conditions,  $fFP = \text{textit}\{\text{fixedFluxPressure}\}\$  corresponds to a Neumann boundary condition for the pressure gradient and hp for Dirichlet boundary condition using the hydrostatic pressure. »

- Figure 2: Please clarify (e.g. in the legend) which is the numerical solution from Aussillous et al. (2013).

The legends have been modified !

- Page 27 line 26: "SUMA" etc does not match up with the labelling in table 9.

The labels, captions and legend have been checked and are now consistent throughout the paper!

- Figure 5: The schematic is quite low resolution -- can this be improved?

The sketch has been completely remade with higher quality.

- Page 31 line 12: esi should be capitalised.

Done

- Page 35 lines 15-16: The final sentence could be removed, or at least avoid use of the first person.

The sentence has been removed.

- Tables 12, 13, and 14: Please improve the captions. The title "Greek Symbols" does not appear to be required.

The captions have been improved.

Consistent with your response to the initial revision, please provide a link to a single archived version of the code on ZENODO.

The zenodo DOI was given and we have now added the full link : https://zenodo.org/record/836643#.Wc47Yoo690s

[revised manuscript text omitted]
[\mathsf{A}^{\mathsf{b}}\right] \cdot \mathbf{u}^{\mathsf{b}} = \mathbf{H}^{\mathsf{b}} + \mathbf{R}^{\mathsf{b}} - \frac{1}{\rho^{b}} \nabla p \tag{51}$$

The matrix  $[A^b]$  is composed of the diagonal terms of the algebraic system associated with equation (50), whereas  $H^b$ 20 includes the off-diagonal terms and the source terms.  $R^b$  is composed of the explicit drag term, the turbulent suspension term, the gravity term and the explicit diffusion terms:

$$\mathbf{R}^{\mathbf{b}} = \frac{\alpha K}{\rho^{b}} \mathbf{u}^{\mathbf{a}} + \mathbf{g} + \frac{\mathbf{f}}{\rho^{b}} \pm \frac{1}{\beta} \nabla \cdot \left\{ \beta \,\nu_{Eff}^{b} \left[ \nabla \mathbf{u}^{bT} - \frac{2}{3} \nabla \cdot \mathbf{u}^{b} \right] \right\} + \frac{1}{\sigma_{c}} \frac{K \,\nu^{bt}}{\rho^{b}} \nabla \alpha \tag{52}$$

The same process can be carried out for the solid phase momentum equation (3) that leads to:

$$\frac{\partial \mathbf{u}^{a}}{\partial t} + \nabla \cdot (\mathbf{u}^{\mathbf{a}} \mathbf{u}^{\mathbf{a}}) - (\nabla \cdot \mathbf{u}^{\mathbf{a}}) \mathbf{u}^{\mathbf{a}} - \frac{1}{\tilde{\alpha}} \nabla \cdot (\nu_{Fr}^{a} \nabla \mathbf{u}^{a}) - \nabla \cdot \left(\nu_{Eff}^{a} \nabla \mathbf{u}^{a}\right) - \nu_{Eff}^{a} \frac{\nabla \alpha}{\tilde{\alpha}} \nabla \mathbf{u}^{a} + \frac{\beta K}{\rho^{a}} \mathbf{u}^{\mathbf{a}} = -\frac{1}{\tilde{\alpha}\rho^{a}} \nabla p^{ff} - \frac{1}{\rho^{a}} \nabla p^{ff}$$

$$+\frac{\beta K}{\rho^{a}}\mathbf{u}^{\mathbf{b}}-\frac{1}{\sigma_{c}}\frac{\beta K\,\nu^{bt}}{\tilde{\alpha}\rho^{a}}\nabla\alpha+\mathbf{g}+\frac{\mathbf{f}}{\rho^{a}}-\frac{1}{\tilde{\alpha}\rho^{a}}\nabla p^{a}+\frac{1}{\tilde{\alpha}}\left\{\nabla\cdot\left[(\alpha\nu_{Eff}^{a}+\nu_{Fr}^{a})\nabla\mathbf{u}^{a}\,^{T}\right]+\nabla\left[\left(\lambda-\frac{2}{3}(\alpha\nu_{Eff}^{a}+\nu_{Fr}^{a})\right)\nabla\cdot\mathbf{u}^{a}\right]\right\}$$

where  $\alpha$  at the denominator is substituted by  $\tilde{\alpha} = \alpha + \alpha_{Small}$  to avoid dividing by zero when the solid phase volume concentration vanishes. This partial differential system of equations can also be written as a matrix equation as follows:

(53)

$$\left[\mathsf{A}^{\mathsf{a}}\right] \cdot \mathbf{u}^{\mathsf{a}} = \mathbf{H}^{\mathsf{a}} + \mathbf{R}^{\mathsf{a}} - \frac{1}{\rho^{a}} \nabla p, \tag{54}$$

5 where the source term  $\mathbf{R}^{\mathbf{a}}$  contains the following terms:

$$\mathbf{R}^{\mathbf{a}} = \frac{\beta K}{\rho^{a}} \mathbf{u}^{\mathbf{b}} - \frac{1}{\sigma_{c}} \frac{\beta K \nu^{bt}}{\tilde{\alpha} \rho^{a}} \nabla \alpha + \mathbf{g} + \frac{\mathbf{f}}{\rho^{a}} - \frac{1}{\tilde{\alpha} \rho^{a}} \nabla p^{a} + \frac{1}{\tilde{\alpha}} \left\{ \nabla \cdot \left[ (\alpha \nu_{Eff}^{a} + \nu_{Fr}^{a}) \nabla \mathbf{u}^{a}^{T} \right] + \nabla \left[ \left( \lambda - \frac{2}{3} (\alpha \nu_{Eff}^{a} + \nu_{Fr}^{a}) \right) \nabla \cdot \mathbf{u}^{a} \right] \right\}$$

$$\tag{55}$$

Following Rusche (2002) the terms involving the ratio of particle phase volume gradient to the volume concentration are treated at the cell face level in the predictor-corrector algorithm. However, we noted the exception of the particle phase normal stress  $p^{ff}$  gradient for which a reconstruction of the surface normal gradient at the cell centre allows to get more stable solutions.

The advantage of separating the right hand side of the momentum equations as the sum of two terms:  $\mathbf{R}$  and  $\mathbf{H}$ , is for writing the pressure-velocity algorithm. A similar method as Rhie and Chow (1983) one can be applied for the gradient terms. The details of the velocity-pressure algorithm are presented in the next subsection.

**3.1 Velocity-pressure algorithm**

10

15 The pressure-velocity coupling and the consequent oscillations in the pressure fields are resolved using the Rhie and Chow method (Rhie and Chow, 1983). The PISO algorithm is used to solve fluid and particle velocities (Rusche, 2002; Weller, 2002; Peltola, 2009).

First, the intermediate velocities  $(\mathbf{u}^{\mathbf{a}*}, \mathbf{u}^{\mathbf{b}*})$  are computed using the corresponding momentum equations (equations without the pressure gradient term:

$$\mathbf{u}^{\mathbf{a}*} = \left[\mathsf{A}^{\mathsf{a}}\right]^{-1} \mathbf{H}^{\mathbf{a}},$$
20
$$\mathbf{u}^{\mathbf{b}*} = \left[\mathsf{A}^{\mathsf{b}}\right]^{-1} \mathbf{H}^{\mathsf{b}},$$
(56)

where  $[A^a]^{-1}$  and  $[A^b]^{-1}$  represent the inverse matrices of  $[A^a]$  and  $[A^b]$ , 
[revised manuscript text omitted]

---

## Author Response (AR3)

**Topical Editor Decision: Publish subject to technical corrections** (20 Oct 2017) by James R. Maddison

Comments to the Author:
Please address the reference to "Appendix 1" on page 11 -- does this refer to an external reference? This could also be clarified at the top of page 12.

The appendix is the notations list at the end of the manuscript. We have modified the references to this appendix in the text on page 11 and 12 as :

*« (see notations list in appendix \ref{notations}) »*

We hope this is clear enough, don't hesitate to let us know if this is not the case.